# Developmental and housekeeping transcriptional programs display distinct modes of enhancer-enhancer cooperativity in *Drosophila*

Vincent Loubiere [1], Bernardo P. de Almeida[1,2], Michaela Pagani[1] & Alexander Stark [1,3] ✉

Genomic enhancers are key transcriptional regulators which, upon the binding of sequence-specific transcription factors, activate their cognate target promoters. Although enhancers have been extensively studied in isolation, a substantial number of genes have more than one simultaneously active enhancer, and it remains unclear how these cooperate to regulate transcription. Using *Drosophila melanogaster* S2 cells as a model, we assay the activities of more than a thousand individual enhancers and about a million enhancer pairs toward housekeeping and developmental core promoters with STARR-seq. We report that housekeeping and developmental enhancers show distinct modes of enhancer-enhancer cooperativity: while housekeeping enhancers are additive such that their combined activity mirrors the sum of their individual activities, developmental enhancers are super-additive and combine multiplicatively. Super-additivity between developmental enhancers is promiscuous and neither depends on the enhancers' endogenous genomic contexts nor on specific transcription factor motif signatures. However, it can be further boosted by Twist and Trl motifs and saturates for the highest levels of enhancer activity. These results have important implications for our understanding of gene regulation in complex multi-enhancer developmental loci and genomically clustered housekeeping genes, providing a rationale to interpret the transcriptional impact of non-coding mutations at different loci.

A key goal in biology is to understand how gene transcription is regulated, as it represents the first step needed for a gene to exert its biological function. This task has proven difficult due to the complexity of gene cis-regulatory landscapes, which typically encompass several discrete regulatory elements, termed enhancers, that jointly shape the activity of their target gene's cognate core promoter[1–3] (CP) and thus gene transcription. Adding to this complexity, cell-type specific developmental genes and housekeeping genes are regulated via two distinct transcriptional programs in *Drosophila*, and their transcription relies on different transcription factors[4] (TFs) and co-factors[5] (COFs).

In the past years, the question of how several concomitantly active enhancers cooperate to drive transcription received increasing attention[6–8]. In other terms, do different enhancers that are each

[1]Research Institute of Molecular Pathology (IMP), Vienna BioCenter (VBC), Vienna, Austria. [2]Vienna BioCenter PhD Program, Doctoral School of the University of Vienna and Medical University of Vienna, Vienna, Austria. [3]Medical University of Vienna, Vienna BioCenter (VBC), Vienna, Austria. ✉e-mail: stark@starklab.org

individually active combine their gene-regulatory functions additively, super-additively or sub-additively toward their target CP? This question is essential because non-coding mutations affecting super-additive enhancers can have an oversized impact on transcription in situ and are therefore more likely to cause downstream functional defects and/or diseases[6]. However, sparse attempts to understand how enhancer-enhancer cooperativity shapes transcription yielded inconsistent outcomes, while multi-enhancer loci are common in both flies[1] and mammals[2].

Early studies suggested that enhancers are additive[7–9], meaning that their combined transcriptional outcome mirrors the sum of their individual activities in linear gene expression space, i.e., the number of produced RNA molecules adds up. However, super-additive[6,8,10–12] and sub-additive modes[8,13] have also been reported, whereby combined enhancers are either stronger or weaker than their summed activities, respectively. For example, *knirps* enhancers have been shown to exhibit additive or super-additive activities in developing *Drosophila* embryos, while *hunchback* enhancers are sub-additive[8]. In mammals, super-additive enhancers were found to be over-represented at cell-type-specific loci[12], and their function was proposed to rely on the formation of COF condensates[6]. Nevertheless, these observations were either inferred from a limited number of enhancer combinations or by using correlative strategies that did not directly measure the enhancers' individual and combined activities. As such, the relative proportion of additive versus non-additive modes of cooperativity and the gene-regulatory contexts in which they are employed remain unclear, as systematic approaches to quantitatively assess such interactions at high throughput are lacking.

Here, we developed an efficient method to simultaneously assess the activity of many individual enhancers and the corresponding pairwise combinations using a single, internally normalized STARR-seq assay. Using *Drosophila* S2 cells as a model system, we measured the individual activity of more than a thousand candidate sequences—spanning a wide range of enhancer activities—and about a million enhancer-enhancer pairs (which we will refer to as enhancer pairs for simplicity) in a tightly controlled, tractable environment. Our results indicate that developmental enhancers that activate tissue-specific genes are super-additive: the activity of enhancer pairs can be accurately predicted using a simple multiplicative model. Consistently, no specific DNA motif signature was strictly required for super-additive interactions between developmental enhancers, which appeared to be largely promiscuous. This result argues against the existence of particularly potent enhancers or enhancer pairs and suggests a rather flexible DNA motif syntax supporting super-additivity, which likely involves Trl and/or Twist binding motifs that—in our system—further boost super-additivity.

In stark contrast, enhancers that activate housekeeping genes behave additively, i.e., their combined activity corresponds to the sum of their individual activities. This functional difference is associated with a higher fraction of Intrinsically Disordered Regions (IDRs) within developmental TFs, which might support downstream super-additive interactions[6,14], while housekeeping regulation might build on the known propensity of housekeeping genes to cluster along the *Drosophila* genome[15].

## Results

### High-throughput quantitative assessment of enhancer pairs

To tackle the modes of enhancer cooperativity at high throughput, we developed a new approach to simultaneously measure the activity of many individual enhancers as well as all pairwise combinations in a single, tightly controlled STARR-seq[1] assay (Fig. 1a). To achieve this, we designed a pool of 300-bp oligos (249-bp candidate sequences flanked by PCR primer binding sites) containing 850 enhancers covering a wide range of activities in *Drosophila* S2 cells, together with 150 randomly selected control sequences (see "Methods", Supplementary

Data 1, 2). We then developed an efficient fusion PCR-based strategy to systematically fuse these 1000 sequences to the 5' and 3' ends of a transcriptionally inert 300 bp spacer sequence, resulting in 1 million combinations including enhancer/enhancer, enhancer/control, control/enhancer and control/control pairs (see "Methods", Supplementary Fig. 1a and Supplementary Data 3). Then, resulting constructs were cloned downstream of a developmental CP so that the activity of each pair would be reflected by its self-transcription (Fig. 1a). We then followed the UMI-STARR-seq protocol[16] to perform the functional screens in *Drosophila* S2 cells.

Using this approach, we were able to measure the individual activity of 970 and 961 candidate sequences in the 5' and 3' locations, respectively, and the activity of 715,479 pairs (see "Methods" and Supplementary Data 4). STARR-seq biological replicates showed a Pearson's correlation coefficient ($r$) of 0.95, and the inferred activities could be validated quantitatively using luciferase assays ($r = 0.81$), indicating that the method is highly reproducible and robust (Supplementary Fig. 1b, c). Moreover, the individual activities of enhancers in the 5' and 3' locations (inferred using enhancer-control and control-enhancer pairs, respectively) were highly correlated ($r = 0.88$) and agreed well with publicly available STARR-seq data[17] (Fig. 1b), indicating that the increased reporter-transcript length and the spacer sequence do not interfere with enhancer function or STARR-seq processing. Furthermore, enhancer pairs were globally stronger than enhancer-control or control-enhancer pairs (Fig. 1c), indicating that two enhancer sequences typically contribute concurrently to transcriptional activation. Accordingly, the activities of enhancer pairs scaled with the individual activities of the enhancers they contain, whereby the presence of a single enhancer either in the 5' or the 3' location was sufficient to drive transcription and maximum activities were achieved by pairing two strong enhancers (Fig. 1d). Despite the slightly increased activity of individual enhancers in the 3' location compared to the 5' location (Fig. 2b, c), the activities of reciprocal pairs (A/B versus B/A) were overall similar and highly correlated ($r = 0.80$, Fig. 1e), indicating that swapping the two candidate sequences between the 5' and 3' locations had no substantial impact on activity. Together, these data show the robustness of our method and its unique potential to directly measure the activity of many individual enhancers and enhancer pairs at an unprecedented scale.

### Developmental enhancers are super-additive

To tackle how enhancers cooperate in pairs, we aimed to predict the activities of enhancer pairs from the individual activities of the enhancers they contain, using either an additive model or a multiplicative model. An additive model posits that the combined enhancer activity is the sum of the individual enhancer activities, i.e., that the numbers of RNA molecules produced add up. Conversely, the multiplicative model posits that the enhancer activities, and thus the number of RNAs, behave multiplicatively (Fig. 2a). Importantly, the additive model tended to under-predict the observed activities of enhancer pairs (in which both candidate sequences are active enhancers) and yielded a rather modest $R$-squared ($R^2$) of 0.65, indicating that developmental enhancers are super-additive (Fig. 2b). The multiplicative model substantially outperformed the additive one ($R^2 = 0.81$, Fig. 2b), and predicted values were more accurate for 88% of enhancer pairs (Fig. 2c), suggesting that developmental enhancers combine multiplicatively.

To further dissect the relationship between individual and combined activities, we fitted a multiplicative model with interaction term (see "Methods"), which slightly improved the prediction accuracy further (Adjusted $R^2 = 0.83$, Fig. 2d, e and Supplementary Fig. 2a, b). One key asset of such a model is to deliver a set of interpretable and informative coefficients. Here, the intercept of 0.11 indicates that pairs consisting of two inactive sequences generally remain inactive, as expected. On the other hand, 5' and 3' coefficients (of 1.12 and 1.06, respectively) were both similar and close to 1.0, indicating that optimal

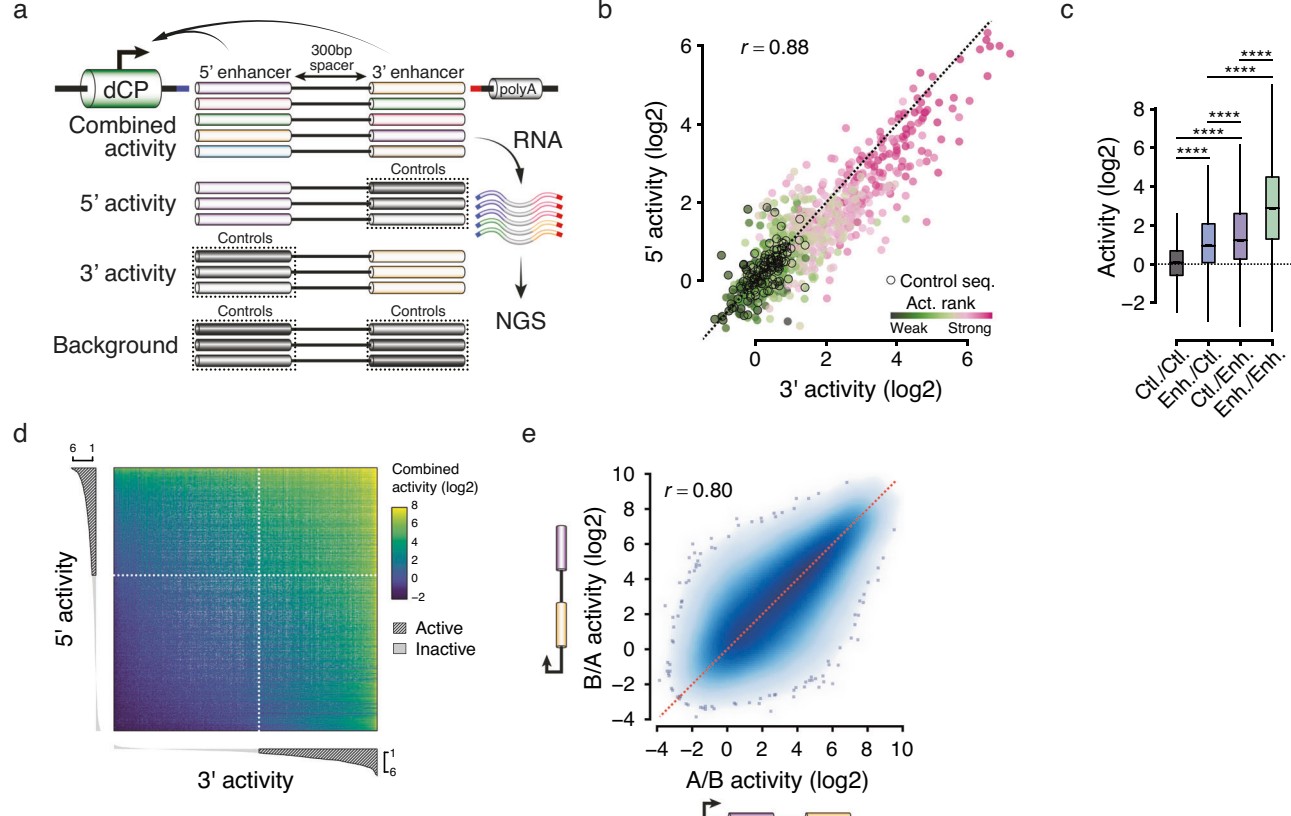

**Fig. 1 | High-throughput assessment of the individual and combined activities of many enhancers. a** Overview of the STARR-seq reporter assay used to simultaneously measure the individual and combined activities of many enhancers. Random control (in gray) and candidate sequences (colors) are fused to the 5′ and the 3′ ends of a transcriptionally inert spacer and cloned downstream of a core promoter, whose transcription mirrors enhancers' individual and combined activities. **b** Correlation between 3′ (x-axis) and 5′ (y-axis) individual activities of 953 candidate sequences. The dotted line represents the identity line (y = x), and Pearson's correlation coefficient is shown on the top left (r). The color code displays sequences' activity rank inferred from a previously published STARR-seq dataset[17]. **c** Quantification of the activity of pairs consisting of two random control sequences (Ctl./Ctl., in gray), one control sequence paired with a candidate sequence either in the 5′ (Enh./Ctl., in blue) or the 3′ (Ctl./Enh., in purple) location, or two enhancer sequences (Enh./Enh., in green). n = 16,032; 90,559; 91,817 and 51,7071 pairs, respectively. Two-sided Wilcoxon test P-values are shown; ****P < 2.2e-308. Box plots show the median (line), upper and lower quartiles (box) ±1.5× interquartile range (whiskers), outliers are not shown. **d** Heatmap of paired activities (see color legend) ranked by individual activities of the 3′ (x-axis) and 5′ (y-axis) candidate sequences. 3′ and 5′ activities are depicted as bar charts on the x and y axes, respectively, with active sequences being highlighted with dashed lines (log2 individual activity >1). **e** Correlation between candidate sequence pairs (A/B, x-axis) and the reciprocal combinations (B/A, y-axis).

predictions were achieved by assuming that both enhancers similarly contribute to the activity of the pair, each to an extent that mirrors its individual activity. Hence, these coefficients substantially agree with a simple multiplicative model.

Interestingly, this refined model revealed a significant, negative interaction between the individual activities of the two enhancers (coefficient = −0.096) that improved prediction accuracy in the highest activity range, for which the simple multiplicative model tended to over-predict (Supplementary Fig. 2c). This presumably indicates that the CP saturates in the presence of very strong enhancers, a phenomenon which has already been shown to constrain enhancer-promoter function[18]. Consistently, the activities of the strongest 5′ or 3′ enhancers can hardly be increased by the addition of a second enhancer (Supplementary Fig. 2d, e).

Altogether, these data suggest a rather simple model, whereby developmental enhancers are super-additive and combine multiplicatively until saturating their cognate CP. Of note, this super-additive behavior was preserved when increasing the spacer size to 2 kb, slightly above the median enhancer-enhancer distance within the *Drosophila* genome (see Supplementary Fig. 2f–h and Supplementary Data 5) and was also validated for ten homotypic enhancer pairs using luciferase assays (Supplementary Fig. 2i). To assess whether this model might be relevant in other cellular contexts, we performed STARR-seq

in S2 cells treated with ecdysone—the major steroid hormone in insects —and in Ovarian Somatic Cells (OSCs), conditions that activate hormone-inducible[19] or OSC-specific[1,4] enhancers, respectively (Supplementary Fig. 2j and Supplementary Data 6, 7). Importantly, these two additional types of developmental enhancers also combined super-additively (Fig. 2f, g, Supplementary Fig. 2k, l), indicating that this is a common feature of *Drosophila* developmental enhancers.

## Developmental enhancer super-additivity is promiscuous

The multiplicative model can accurately predict the activity of enhancer pairs, using only the activities of individual enhancers and no additional information (regarding, for example, the enhancers' sequences or native genomic contexts). Thus, it precludes the existence of large proportions of additive or sub-additive combinations and rather suggests that developmental enhancers are generally super-additive, challenging the existence of complex rules for how they interact with each other. Nevertheless, some pairs remain stronger or weaker than predicted (Figs. 2d, 3a), which might reflect additional rules not considered by the multiplicative model. We thus sought to investigate whether these differences—referred to as "residuals"— could be associated with specific DNA motif signatures since, in the STARR-seq setup, enhancer pairs only differ by their DNA sequences while everything else is kept constant.

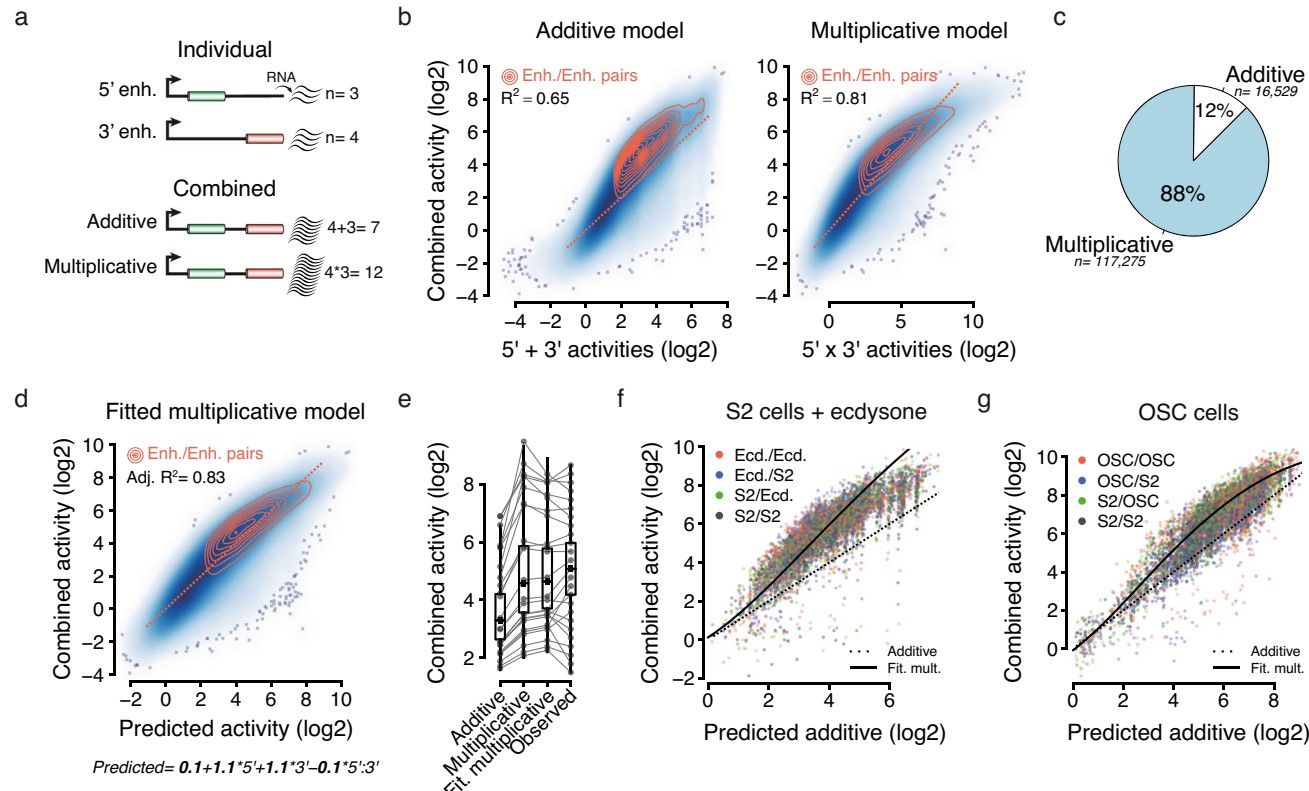

**Fig. 2 | Developmental enhancers combine multiplicatively. a** Schematic illustration of an additive model (where the number of RNA molecules produced by each enhancer add up) and a multiplicative model (where the number of RNAs combine multiplicatively). **b** Scatterplots showing predicted activities (*x*-axis) based on an additive (left) or a multiplicative model (right) versus observed activities (*y*-axis), with corresponding *R*-squared ($R^2$) values. Enhancer pairs in which both candidate sequences are active are highlighted using density lines (in orange), and dotted lines correspond to identity lines ($y = x$). **c** Fraction of enhancer pairs (in which both candidate sequences are active) for which the additive (in white) or the multiplicative (in blue) predicted values were the most accurate. **d** Fitted multiplicative model with interaction term using the 5′ and 3′ individual activities to predict the activities of enhancer pairs, with the corresponding adjusted $R^2$ value (top left). Enhancer pairs in which both candidate sequences are

active are highlighted using density lines (in orange). Dotted lines correspond to the identity line ($y = x$). Fitted coefficients and resulting equation are shown at the bottom. **e** Box plots showing, for all enhancer pairs, predicted values using the three different models and observed values (*x*-axis). Lines connect the values for a representative set of pairs, spanning the dynamic range of observed activities. $n = 517{,}071$ pairs per box plot; box plots show the median (line), upper and lower quartiles (box) ±1.5× interquartile range (whiskers), outliers are not shown. **f**, **g** Predicted additive (*x*-axis) versus observed combined activities for ecdysone-inducible enhancers and OSC-specific enhancers (in red) in ecdysone-treated S2 cells (**f**) or OSC cells (**g**). As a reference, a subset of pairs containing enhancers that were also active in S2 cells are shown (see color legend). Dotted lines depict the identity lines (where observed combined activities equal expected additive outcomes), and solid lines represent fitted multiplicative models.

We therefore systematically measured the association of a large collection of TF motifs with the enhancer activities and with the residuals, the latter reflecting the difference between the activities expected according to the multiplicative model and the observed activities (see "Methods" and Supplementary Data 8). In line with their prominent role in driving enhancer activity in S2 cells[17,20], homotypic and heterotypic combinations of the AP-1 and GATA motifs (either in the 5′ or 3′ enhancers) were associated with stronger overall enhancer activity, but not with increased or decreased residuals (Fig. 3b). Compared to the range of activities, the range of residuals is about an order of magnitude smaller (between −0.3 and +0.3 versus −1 to +2, each on a log2 scale; Fig. 3b). This is consistent with the accuracy of the activity-based multiplicative model and argues that super-additivity between developmental enhancers is promiscuous in S2 cells, with motif syntax rules having only minor influences.

To nevertheless test whether super-additivity might be enhanced by specific TF motifs, we focused on the Trl motif (GAGA) and a variant of the Twist motif (CATATG), which were associated with higher residuals (see Supplementary Fig. 3a) but not with substantially increased overall enhancer activity (Fig. 3b). We selected 50 enhancers containing at least three Trl or two Twist motifs (see PWMs in Supplementary Data 8), mutated the motifs and measured the activity of the resulting

pairs using STARR-seq (Supplementary Data 9, 10). Although mutant enhancer pairs remained super-additive, they showed slightly but significantly decreased super-additivity compared to their wt counterparts (Fig. 3c). Conversely, pasting Trl motifs into enhancers that did not contain any such motif moderately but significantly increased their super-additivity and a similar, albeit weaker trend was observed using Twist motifs (Fig. 3d). Thus, although their motifs are dispensable for super-additivity, Trl and Twist TFs slightly boost such interactions in S2 cells.

Consistent with the absence of a clear association between specific motifs and strongly enhanced/decreased super-additivity, a LASSO regression using motif counts as input performed poorly at predicting the residuals of the multiplicative model in untreated S2 cells, ecdysone-treated S2 cells and OSC cells ($R^2 \approx 0.06 \pm 0.01$ on held-out test sets, see Supplementary Fig. 3b). In contrast, such model performed substantially better at predicting the activity of enhancer pairs in S2 cells ($R^2 = 0.35 \pm 0.04$, Supplementary Fig. 3c), and unambiguously identified the motifs that are known to support the activity of S2 enhancers[17], ecdysone-inducible[19] and OSC-specific enhancers[4] (Supplementary Fig. 3d).

Besides the moderate boost in enhancer super-additivity that Trl and Twist may foster, our results overall indicate that super-additivity

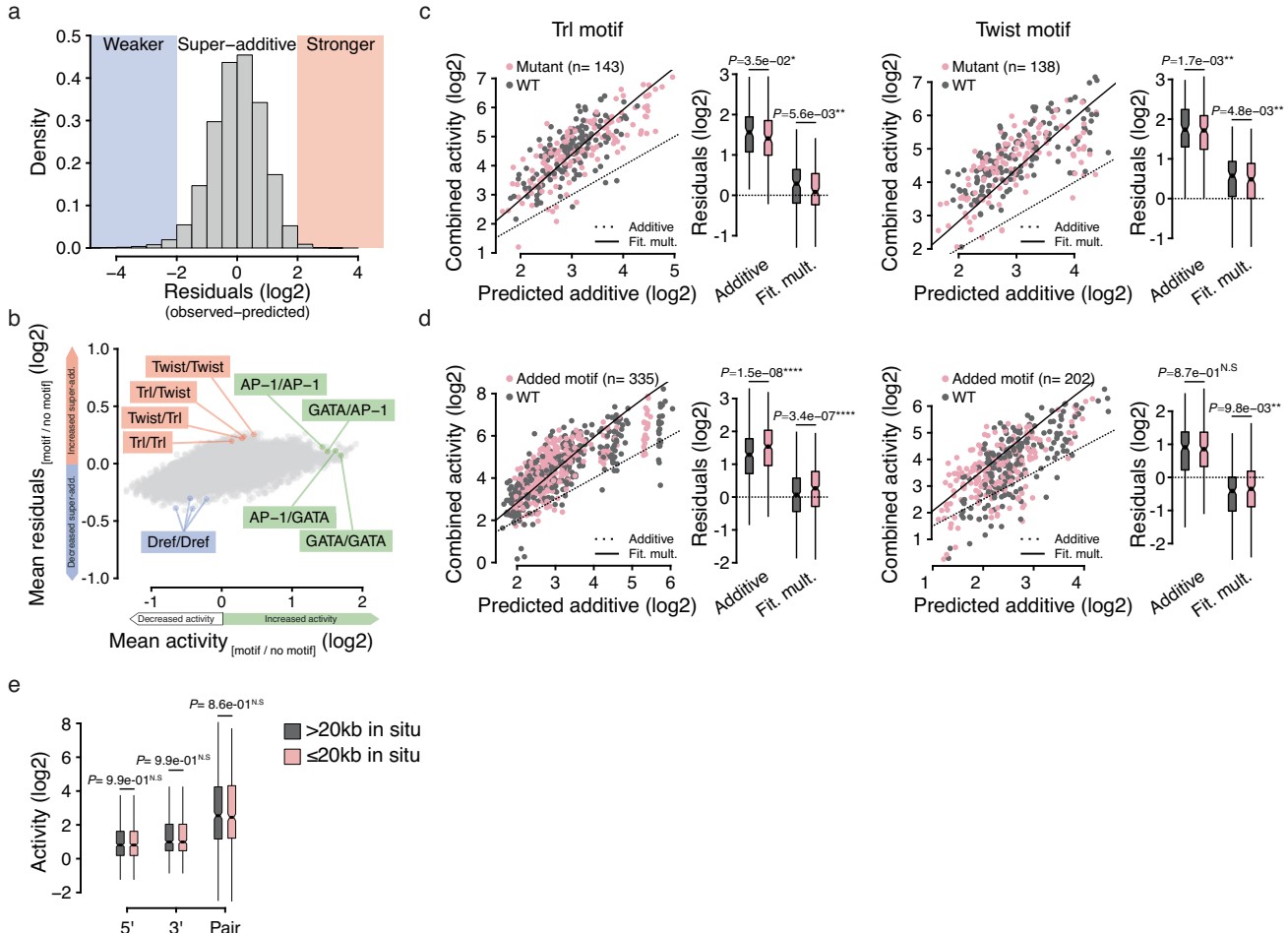

**Fig. 3 | Developmental enhancer super-additivity is promiscuous. a** Distribution of the residuals of the fitted multiplicative model with interaction term, ranging from negative (weaker super-additivity, in blue) to positive (stronger super-additivity, in orange). **b** For each pairwise combination of TF motifs (120 × 120 = 14,400), the mean impact on the activity (x-axis) and on the residuals (y-axis) of corresponding enhancer pairs are shown. For example, Trl/Twist pairs (in which the 5' and 3' enhancers contain at least one instance of the Trl and Twist motifs, respectively) show globally increased residuals (in orange). AP-1 and GATA motifs strongly affected enhancer activity, while the Dref motif negatively impacted activity and super-additivity. **c, d** Impact of mutating (**c**, in pink) or adding (**d**, in pink) Trl (left) or Twist motifs (right) on the predicted additive (x-axis) versus observed combined activities (y-axis) of developmental enhancer pairs. As a reference, corresponding wild-type (WT) developmental enhancer pairs are shown in gray. Dotted lines depict the identity line (where observed combined activities equal expected additive outcomes), and solid lines represent the fitted multiplicative model. For each condition, the residuals of WT versus mutant pairs were quantified (see box plots on the right) using either the additive or the fitted multiplicative model (Fit. mult, see x-axis) and compared using paired, two-sided Wilcoxon tests. Box plots show the median (line), upper and lower quartiles (box) ±1.5× interquartile range (whiskers), outliers are not shown. **e** Quantitative comparison of the activity of enhancer pairs from the same locus (≤20 kb distance in situ, in pink) versus an activity-matched set of distant enhancer pairs (>20 kb, in gray). n = 567 pairs per box plot. Two-sided Wilcoxon test P-values are shown; box plots show the median (line), upper and lower quartiles (box) ±1.5× interquartile range (whiskers), outliers are not shown.

between developmental enhancers does not rely on rigid motif syntax rules nor on a specific TF or a defined combination of TFs and—in our system—is largely promiscuous. To test whether enhancer pairs that are found in the same locus (≤20 kb apart) and could cooperate in situ might have evolved to enhance their super-additivity (via means that would not be captured by classical motif analyses), we compared such pairs to a control set of more distant enhancers (>20 kb in situ), and found no substantial difference in our system (Fig. 3e).

### Housekeeping enhancers are additive

In *Drosophila*, tissue-specific developmental genes and housekeeping genes form two different transcriptional programs that rely on distinct sets of enhancers, CPs and TFs[4,5]. Dref is a key regulator of housekeeping genes in *Drosophila*[5,21] and, interestingly, its DNA binding motif was associated with substantially lower residuals (Fig. 3b), suggesting that it might impair the super-additivity of developmental enhancers. Consistently, pasting Dref motifs within developmental

enhancer pairs significantly reduced their residuals but also the individual activity of each enhancer (Supplementary Fig. 3e, f). Hence, it is unclear whether this reduced super-additivity is enhancer-intrinsic or related to the developmental CP, which cannot be efficiently activated by housekeeping-type enhancers and TFs.

To address this question, we decided to assess the activities of enhancers and enhancer pairs toward the RpS12 housekeeping CP. We designed a smaller synthetic DNA library containing 62 housekeeping enhancers, 53 developmental enhancers and 50 control sequences (Supplementary Data 11), paired all sequences systematically as above, and cloned all pairs downstream of the RpS12 housekeeping CP. We then performed STARR-seq to measure the activities of the individual enhancers and enhancer pairs as described[16] and modeled the data with the additive and multiplicative models (see Supplementary Data 12). In contrast to the previous developmental setup, the additive model performed significantly better at predicting the activities of housekeeping enhancer pairs ($R^2 = 0.31$) than a simple multiplicative

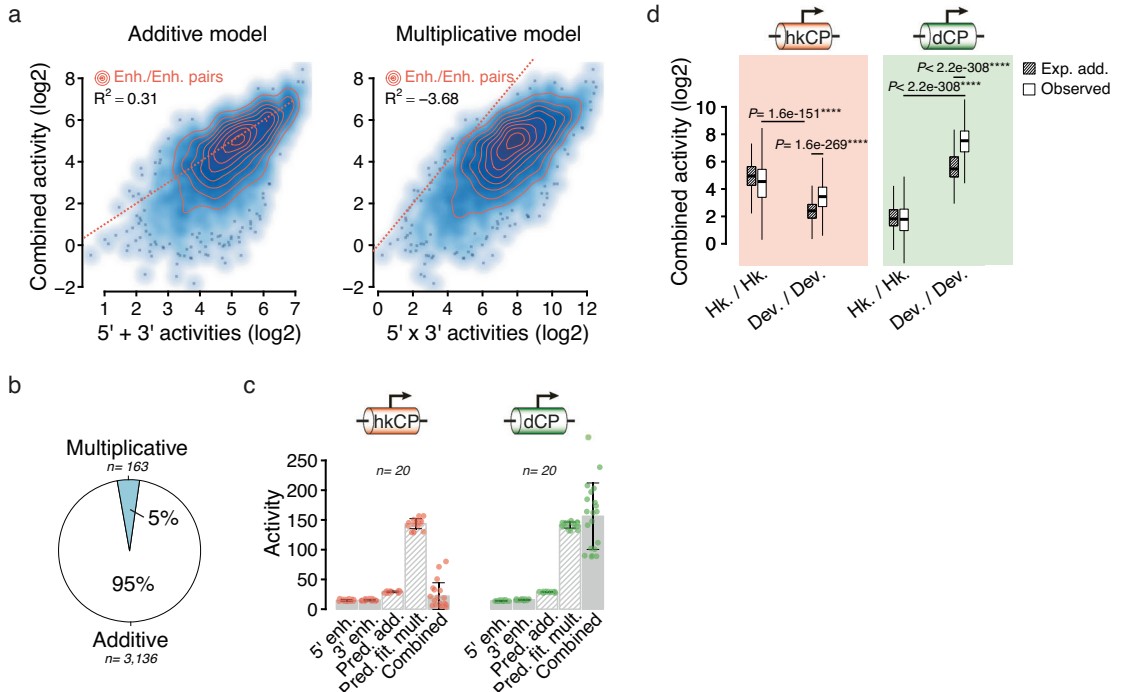

**Fig. 4 | Housekeeping enhancers are additive and developmental enhancers are super-additive independently of the CP type. a** Scatterplots showing predicted activities (*x*-axis) based on an additive (left) or a multiplicative model (right) versus observed activities (*y*-axis) using the RpS12 housekeeping CP. Corresponding *R*-squared (*R*²) values are shown (top left), and enhancer pairs in which both candidate sequences are active are highlighted using density lines (in orange). Identity lines are shown using dotted lines (*x* = *y*). **b** Fraction of enhancer pairs (in which both candidate sequences are active) for which the additive (in white) or the multiplicative (in blue) predicted values were the most accurate. **c** Selected housekeeping (left) and developmental (right) enhancer pairs with comparable 5′ and 3′ individual activities, either with a housekeeping (hkCP, in red) or a developmental (dCP, in green) Core Promoter. For each pair, individual and combined measured activities are shown (solid gray bars) and compared to predicted activities (striped bars) using either the additive (Pred. add.) or the fitted multiplicative (Pred. fit. mult.) model. Bar heights correspond to the mean activity values and whiskers to the standard deviations. **d** Expected additive and observed activities of housekeeping versus developmental enhancer pairs (*x*-axis) using either a housekeeping (hkCP, in red) or a developmental (dCP, in green) CP. *n* = 3590, 2392, 3513 and 2328 pairs, respectively. Two-sided Wilcoxon test *P*-values are shown; box plots show the median (line), upper and lower quartiles (box) ±1.5× interquartile range (whiskers), outliers are not shown.

model (*R*² = −3.68) and was more accurate in 95% of the cases (Fig. 4a, b). Hence, housekeeping enhancers interact additively, even in the presence of a compatible CP.

To further compare developmental and housekeeping contexts side-by-side, we also measured the activity of this smaller library using the DSCP developmental CP (see Supplementary Data 13). In line with the absence of super-additive interactions between them, housekeeping enhancer pairs showed substantially lower activity levels compared to developmental enhancer pairs with comparable 5′ and 3′ individual activities (Fig. 4c). Nevertheless, the known specificity between developmental and housekeeping enhancer-CP[4] holds true: with the housekeeping CP, maximum activities are achieved by combining two housekeeping enhancers, while the developmental CP reaches its maximum levels with developmental enhancer pairs (Fig. 4d). Importantly however, the developmental enhancers are super-additive and the housekeeping enhancers additive irrespective of the CP type, indicating that these distinct modes of cooperativity are enhancer-intrinsic properties independent of the CP (Fig. 4d).

Together, our results indicate that housekeeping and developmental enhancers are intrinsically different in their modes of cooperativity, namely additive versus super-additive, respectively, and that this inherent distinction is independent of their interaction with the CP.

## Discussion
Here, we developed a new approach to study enhancer-enhancer cooperativity at an unprecedented scale, uncovering an unexpected

discrepancy between developmental and housekeeping transcriptional programs in *Drosophila*. While developmental enhancers activating tissue-specific genes are super-additive in our minimal reporter assay, housekeeping enhancers behave additively. Further dissection of the developmental dataset suggests a rather simple model, where the activity of the two enhancers combine multiplicatively until they eventually saturate the CP. Thus, super-additive interactions between regulatory elements might be more widespread than previously thought, with a recent study showing that enhancer and promoter activities multiplicatively combine to determine RNA output in mammals[18].

Although STARR-seq in cultured cell lines does not capture all the aspects of enhancer cooperativity during development, super-additivity between developmental enhancers might explain the predominance of developmental genes among genes with the highest transcription rates as measured by PRO-Seq[22] (Supplementary Fig. 3g) and enable rapid gene induction after signaling and during development. It also has important implications for mutations, since a single mutation affecting a single super-additive enhancer might have a drastic effect on transcription[8] and potentially influence disease risk[6]. On the other hand, enhancer super-additivity plus promoter saturation might also foster the known robustness of developmental loci containing many enhancers[2]: given our finding that CP saturation constrains the transcriptional outcome of pairs of strong enhancers, even the full mutational disruption of one enhancer would have no impact as the remaining enhancers would be sufficient to drive maximal transcription. At such complex loci, removing one or several

enhancers has indeed no impact on transcription[2] (i.e., enhancers act redundantly), implying that the combined activity of enhancers initially exceeds the capacity of the promoter, and future studies should aim at systematically measuring the saturation levels of various CPs.

Interestingly, while Trl and Twist motifs seem to positively influence super-additivity in our setup, we did not find specific DNA motif signatures that are strictly required for super-additive interactions between developmental enhancers, arguing against the existence of strong specificities between subsets of developmental enhancers. Thus, the way TFs and COFs interact to dictate transcription might slightly differ when considering enhancer pairs versus single enhancers. In a single developmental enhancer, distinct combinations of TFs were shown to exhibit distinct types of additive and cooperative behaviors[17,23,24]. In contrast, our results suggest that higher-order interactions between the ensemble of TFs and COFs that each developmental enhancer recruits are less specific, since they globally lead to super-additive outcomes, even when adding one developmental enhancer to housekeeping enhancer-CP pairs (Supplementary Fig. 3h). However, the *Drosophila* genomic enhancer sequences we have been using typically already contain homotypic and heterotypic combinations of motifs, and future studies could use synthetic sequences to more specifically assess the impact of each motif.

In contrast with developmental enhancers, we found housekeeping enhancers to behave additively, suggesting that the molecular mechanisms governing the transcription of *Drosophila* developmental and housekeeping genes are fundamentally different. Notably, a recent study suggested that super-additive/synergistic interactions might indeed be characteristic of developmental loci in mouse, while other enhancers would be additive[12]. Altogether, these findings lead us to postulate that developmental TFs might have specifically evolved the ability to foster super-additive interactions, presumably in order to support sharp transcriptional changes in response to developmental cues. Interestingly, for the different TF motifs within a single enhancer, we confirmed the TF motifs' promoter-selectivity and their multiplicativity—sometimes referred to as motif cooperativity or synergy[17]—for both developmental and housekeeping enhancers (Supplementary Fig. 3i–k). These results indicate that the modes of cooperativity between different housekeeping and developmental enhancers do not reflect the cooperativity of cognate TFs within individual enhancers.

In recent years, extensive investigations have focused on the function of Intrinsically Disordered Domains (IDRs), whose presence in TFs/COFs has been associated with the formation of condensates[14] and super-additive transcriptional outcomes[6]. By comparing TFs and COFs showing a preference for either developmental or housekeeping enhancers[25] (see "Methods"), we found that developmental TFs/COFs contain significantly longer IDRs (Supplementary Fig. 3l), which might favor super-additive interactions at developmental enhancers in *Drosophila*. Consistently, dual enhancers—a subset of housekeeping enhancers that also activate developmental CPs[4] and contain more Trl and Twist developmental-type motifs—combine super-additively toward a housekeeping CP, contrasting with the additive behavior of canonical housekeeping enhancers (Supplementary Fig. 3m, n). However, further studies would be needed to investigate how additive versus super-additive behaviors are encoded at the protein level, and whether other chromatin-related features might further constrain enhancer cooperativity in situ[12], and how motif binding affinity might influence these interactions[8].

Finally, additive interactions seem sufficient to foster steady transcription of housekeeping genes. However, such interactions still imply that housekeeping enhancers might boost each other, which could explain why housekeeping genes and enhancers tend to form clusters along the *Drosophila* genome, an arrangement that has previously been shown to be important for their proper transcription[15]. Future studies should aim at integrating the basic modes of cooperativity between active enhancers that we uncovered here with further

regulatory information (e.g., enhancer-promoter distance, CP selectivity, CP saturation) and chromatin states toward the overarching goal of achieving genome-wide predictions of gene activity.

## Methods
### Design of oligo pools
For this study, we designed three pools of oligonucleotides, consisting of 249-bp candidate sequences flanked by PCR primer binding sites, for a total length of 300nt. For their design, we used publicly available STARR-seq data[4,17,19] and DHS data[1] from *Drosophila* S2 and OSC cells (see the "Data availability" section for corresponding GEO repositories). Unless explicitly mentioned, genomic sequences originated from the dm3 version of the *Drosophila* genome. A summary of the composition of each library is available in Supplementary Data 1. The genomic coordinates and DNA sequences of all oligo pools are available in Supplementary Data 2, 9, 11.

A first pool of 1000 oligos was designed to comprehensively assess how enhancer pairs function downstream of a developmental CP. It contained 600 developmental enhancers, 100 housekeeping enhancers, 150 control sequences and 150 inducible/OSC-specific enhancers[4,19] (see "WT oligo pool", Supplementary Data 1 and 2). Developmental and housekeeping enhancers were selected to cover a wide range of activities in *Drosophila* S2 cells. For control sequences, we randomly sampled 100 genomic sequences showing no STARR-seq signal in S2 cells, 25 exon sequences and 25 sequences from the 20080805 version of the *E. coli* genome.

To test the impact of Trl, Twist and Dref motifs on cooperativity, we designed a pool of 465 WT sequences and 533 mutated variants (998 sequences in total). WT sequences contained 93 randomly sampled inactive genomic sequences showing no STARR-seq signal in S2 cells, 106 DHS sites showing no STARR-seq signal in S2 cells, 131 developmental enhancers containing no Twist or Trl binding motifs, 50 developmental enhancers containing at least two Twist motifs, 50 developmental enhancers containing at least three Trl motifs and 35 enhancers showing both developmental and housekeeping activities (termed "shared" enhancers) containing at least two Dref motifs (see the "Mutated oligo pool" in Supplementary Data 1). The position weight matrices used to identify instances of the Twist/Trl/Dref motifs are available in Supplementary Data 8.

To assess the relevance of Twist/Trl/Dref motifs, we pasted these motifs into sequences with no STARR-seq activity or in active developmental enhancers that did not contain any such motifs. Conversely, we mutated Twist/Trl/Dref motifs in a subset of active enhancers that contained them, by replacing each motif instance with randomly sampled stretches of nucleotides (containing no known motifs). Because we are specifically interested in cooperativity between active enhancer pairs, we needed to avoid mutations that would alter the activity of the individual enhancers by, for example, creating new motifs and/or deleting essential ones, as this would confound the analysis. In other words, our goal was to preserve the activity of the individual enhancers, while changing the TF motifs we suspected to influence cooperativity. To do so, we started from a large pool of WT sequences and generated 1000 possible enhancer variants for each condition, changing the position of pasted motifs and/or the stretches of nucleotides being used to replace each motif instance. Then, we predicted the activity of all enhancer variants using DeepSTARR[17] and retained the ones with the smallest impact on predicted individual activities (for each tested condition, the final number of variants can be found in Supplementary Data 1, and the full sequences of WT and mutated variants are available in Supplementary Data 9).

For the side-by-side comparison of housekeeping and developmental enhancer-enhancer pairs with different CPs, we designed a pool containing 165 candidate sequences: 50 randomly samples inactive sequences showing no STARR-seq signal, 62 housekeeping and 53 developmental enhancers and 9 shared enhancers, which can activate

both developmental and housekeeping CPs (see "Focused oligo pool", Supplementary Data 1 and 11).

## Synthesis of enhancer-enhancer pairs

300-mer oligo pools were synthesized by Twist Bioscience, amplified and split into two batches that were processed in parallel to generate 5' and 3' candidate sequences. Using overhang PCRs, Illumina adapters and CG-rich overhangs were added to the ends of 5' and 3' candidates (see Supplementary Fig. 1a and Supplementary Data 3 for a schematic view of the method and corresponding PCR primers). For the library containing a 2 kb spacer, a MaubI restriction site was also added upstream of the 5' candidate sequences (see Supplementary Data 3). Transcriptionally inert spacers were amplified from *Drosophila* genomic DNA (dm3 genomic coordinates: chr2L:509283-509549 (300 bp spacer); chr2R:14136948-14138782) and flanked with complementary CG-rich overhangs.

This way, the three DNA fragments (5' candidate, spacer, 3' candidate) contain overlapping sequences at their extremities, corresponding to CG-rich overhangs that were optimized to allow efficient, orientation-specific fusion PCR reactions[26] (see Supplementary Fig. 1a). Following 20 cycles of linear amplification, fused fragments were amplified for 20 PCR cycles and flanked with overhang sequences compatible with Gibson Assembly® (see primers in Supplementary Data 3). Finally, fused enhancer pairs were size selected using gel purification (-1 kb for the 300 bp spacer, 2.6 kb for the 2 kb spacer).

## STARR-seq library cloning and sequencing

STARR-seq libraries were generated by cloning enhancer pairs into the *Drosophila* STARR-seq vectors, containing either the developmental *Drosophila* synthetic core promoter[27] (DSCP) or the housekeeping RpS12 core promoter[4]. We followed the previously established UMI-STARR-seq library cloning protocol[16], except that the In-Fusion HD reaction was replaced by Gibson assembly® (New England BioLabs, #E2611S). Briefly, two Gibson Assembly® reactions were used for each library (500 ng of digested plasmid, three molar excess of enhancer pair constructs and 40 µL 2X Gibson Master Mix, for a total volume of 80 µL per reaction) following the manufacturer's instructions. Assembled libraries were electroporated into competent *E. coli* (Invitrogen, #C640003) and grown O/N in 4 L LB-Amp (Luria-Bertani medium plus ampicillin, 100 µg/mL) and purified using Plasmid Plus Giga Kit (Qiagen, #12991). Then, libraries containing a 300 bp spacer were UMI-tagged and amplified as previously described[16].

For the 2 kb spacer library, a second MaubI restriction site was first added at the 3' end of enhancer pairs, using 200 ng of plasmid and 2 PCR cycles (2X HiFi HotStart ReadyMix (Roche #07958927001), forward primer: TGTACAACTGATCTAGAGCATGCA, reverse primer: GATAATCCGCGCGCGCTCATCAATGTATCTTATCATGTCTG. Program: 98 °C 45 s, (98 °C 15 s, 65 °C 30 s, 72 °C 120 s) × 2 cycles, 72 °C 180 s). To excise enhancer pairs flanked by MaubI restriction sites and get rid of the methylated plasmid template, products were digested for 2 h at 37 °C with MaubI and DpnI enzymes (ThermoFisher Scientific # FD2084 and # FD1703) and purified, before O/N ligation at 16 °C (100 µL DNA template, 120 µL T4 buffer, 0.8 µL High Concentration T4 DNA Ligase (New England Biolabs # M0202M)). After inactivation (20 min at 65 °C), 1 µL of NotI enzyme (ThermoFisher Scientific # FD0593) was directly added to the reaction and incubated for 2 h at 37 °C, in order to re-linearize ligated products, which were finally UMI-tagged and amplified as previously described[16].

Finally, all libraries were sequenced at the VBCF NGS facility using Next-generation Illumina sequencing, following the manufacturer's protocol with standard Illumina i5 indexes and UMIs at the i7 index (paired-end 36nt or longer for WT oligo pools, paired-end 150nt for the mutated oligo pool).

## STARR-seq screens

*Drosophila* cells were cultured at 27 °C and passaged every 2–3 days. S2 cells (obtained from ThermoFisher Scientific, #R69007) were cultured in Schneider's *Drosophila* Medium (SM, Gibco, #21720-024) supplemented with 10% heat-inactivated FBS (Sigma, #F7524) and 1% Penicillin/Streptomycin. OSC cells (obtained from DGRC, stock #288) were cultured in M3 Insect Medium supplemented with 0.6 mg mL-1 glutathione, 10% FBS, 10 mU mL-1 insulin and 10% fly extract. For each biological replicate, 10^8 cells were resuspended in 400 µL of a 1:1 dilution of HyClone MaxCyte electroporation buffer and serum-free SM and electroporated with 20 µg of the input libraries (see previous section) using the MaxCyte-STX system ('Optimization 1' protocol). For the induction of hormone-inducible enhancers, S2 cells were treated after 1 h of recovery following electroporation, by adding 50 µL of 10 mg/mL with 20-Hydroxyecysone to 25 mL flasks (-42 µM final concentration). Electroporated cells were then collected after 24 h, and libraries containing a 300 bp spacer were processed following the established UMI-STARR-seq protocol[16] (of note, PCR elongation time was increased to 55 s). For the library containing the 2 kb spacer, a specific primer was used for reverse transcription (GATAATCCGCGCGCGCTCATCAATGTATCTTATCATGTCTG), which adds a MaubI restriction site at the 3' end of the construct (see Supplementary Fig. 2g for a schematic view of the method). Then, the usual second-strand PCR step was replaced by 20 cycles of linear amplification (program: 98 °C 45 s, (98 °C 15 s, 65 °C 30 s, 72 °C 150 s) × 20 cycles). To excise enhancer pairs flanked by MaubI restriction sites and get rid of the methylated plasmid template, products were digested for 2 h at 37 °C with MaubI and DpnI enzymes (ThermoFisher Scientific # FD2084 and # FD1703) and purified, before O/N ligation at 16 °C (100 µL DNA template, 120 µL T4 buffer, 0.8 µL High Concentration T4 DNA Ligase (New England Biolabs # M0202M)). After inactivation (20 min at 65 °C), 1 µL of NotI enzyme (ThermoFisher Scientific # FD0593) was directly added to the reaction and incubated for 2 h at 37 °C, in order to re-linearize ligated products. UMI-tagging, nested-PCR and sequencing-ready PCR steps were then performed as previously described[16].

Finally, sequencing-ready libraries were size selected on a 1% agarose gel (-1 kb fragments for the 300 bp spacer, 600 bp for the inverse-PCR products resulting from the ligation of the 2 kb spacer, see Supplementary Fig. 2g) and sequenced at the VBCF NGS facility using Next-generation Illumina sequencing, following the manufacturer's protocol with standard Illumina i5 indexes and UMIs at the i7 index (paired-end 36nt or longer for WT oligo pools, paired-end 150nt for the mutated oligo pool).

## Next-generation sequencing data processing

For each oligo pool, a custom index containing the corresponding sequences was generated using the buildindex function from the Rsubread R package[28] (version 2.12.2). STARR-seq paired-end reads were then aligned using the align function from the same package, with the following parameters: type = "dna", unique = TRUE, maxMismatches = 3 (or maxMismatches = 5 for the 150nt reads of the mutated oligo pool). Only the pairs for which both reads could be aligned with expected orientations and positions were considered. Then, UMI sequences were retrieved and collapsed as previously described[17].

## Downstream analyses

All downstream analyses were performed in R[29] (version 4.2.0) using the data.table package[30] (version 1.14.6). For all *P*-values shown in the figure panels and legends, the following abbreviations were used: *P < 5e-2, **P < 1e-2, ***P < 1e-3, ****P < 1e-5, N.S = not significant.

## Computation of individual and combined activities

For the calculation of individual and combined enhancer activities, only the heterotypic pairs with at least 5 UMI-collapsed read counts in

each of the two input replicates were considered. One pseudo count was added to UMI-collapsed read counts and the log2 fold change over input were computed using DESeq2 (v1.38.3)[31] (with at least two replicates). To normalize the samples between them and facilitate comparisons between different screens, we used the negative control sequence pair counts as scaling factors, so that the activities of negative control pairs are centered on zero.

Before computing the individual activity of each candidate sequence, we first aimed at removing potential outlier negative control sequences, that might eventually show some activity in our screens. To do so, we assessed the activity of each negative control sequence by averaging its activity across all its observed combinations with other control sequences. Resulting activities were scaled and only the 5′ and 3′ control sequences with a z-score value located between −1 and 1 were considered as valid, robust control sequences. Finally, these robust control sequences were used to compute the individual activity of each individual candidate sequence (which we refer to as "enhancers" for simplicity), by averaging their activities across all its observed combinations with at least 10 robust control sequences (otherwise, the individual candidate sequence was discarded). To classify an enhancer as active, we compared the activities of all its observed combinations with robust control sequences to the activities of control/control pairs using one-tailed Fisher's exact tests (alternative = "greater") followed by false discovery rate (FDR) multiple testing correction. Only the enhancer sequences with a log2 activity bigger than 1 and an FDR < 0.05 were considered active. For each STARR-seq screen, individual and combined activities are reported in Supplementary Data 4–7, 10, 12, 13.

## Modeling of combined activities using individual activities

The individual activities of the 5′ and 3′ enhancers were scaled using negative control sequence pairs (of note, all pairs containing a negative control sequence were discarded prior to modeling combined activities). Therefore, they correspond to fold-changes normalized to the basal activity of the core promoter expressed in log2. Hence, for a given pair $P$, log2 additive and multiplicative predicted values were computed using the following formulas:

$$P_{additive} = \log 2(2^A + 2^B - 1) \qquad (1)$$

$$P_{multiplicative} = A + B \qquad (2)$$

Where $A$ and $B$ correspond to the log2 individual activities of the 5′ and 3′ enhancers, respectively. To optimize the performance of the multiplicative model, we fitted the following multiplicative model with interaction term using log2 activity values and the $lm$ function in R:

$$P_{activity} = \beta 0 + \beta 1 * A + \beta 2 * B + \beta 3 * A * B + \epsilon \qquad (3)$$

Where $\beta 0$ is the intercept, $\beta 1$ and $\beta 2$ represent the contribution of each enhancer's activity, $\beta 3$ captures the interaction between the two enhancers and $\epsilon$ is the error term. Unless explicitly stated, multiplicative models were fitted using the entire datasets after removing the pairs containing negative control sequences. The performance of each model was assessed using $R$-squared ($R^2$) computed with the following formula:

$$R^2 = \frac{SS_{regression}}{SS_{total}} \qquad (4)$$

Where $SS_{regression}$ and $SS_{total}$ correspond to the Sum of Squares due to regression and the total Sum of Squares, respectively. For fitted models, $R$-squared values were adjusted (Adj. $R^2$) to account for the number of predictors.

## Saturation analysis

To assess the saturation of the developmental CP, we focused on pairs containing a weak (log2 individual activity between 1 and 1.5) or a strong (log2 individual activity > maximum(individual activity) − 1) developmental enhancer, either in the 5′ or the 3′ location. For these two groups (weak and strong), the mean activity was computed for each second enhancer in the pair.

## DNA binding motifs analyses

To assess the impact of DNA binding motifs on the activity of developmental enhancers and/or super-additive interaction between them, we used a publicly available collection of 13,899 annotated position weight matrices (PWMs) classified in 901 curated, non-redundant clusters[17,32]. Using publicly available RNA-Seq data[1,19], we selected the clusters for which at least one of the related PWMs is associated to a TF that is expressed in *Drosophila* S2 cells, S2 cells treated with ecdysone or OSC cells, resulting in 3310 motifs from 233 clusters. For each PWM, we counted the number of motifs within all the oligonucleotides of the WT library pool (see previous sections, Supplementary Data 2) and a control set of 1000 249-bp control sequences (randomly sampled from the dm3 version of the *Drosophila* genome), using the motif_counts function from the motifmatchr R package[33] (version 1.18.0) with the following parameters: genome = "dm3", bg = "genome", p.cutoff = 5e-04. Using two-tailed fisher tests followed by FDR multiple testing correction, we selected the motifs that were significantly over- or under-represented in any of the groups of candidate sequences from the WT oligo pool (see the "group" column for the WT oligo pool in Supplementary Data 1) compared to the set of control random sequences (FDR < 1e-5) and had at least 5 counts in total; resulting in a curated, non-redundant set of 120 binding motifs (listed in Supplementary Data 8).

To assess the effect of a specific combination of motifs on the activity of developmental enhancer-enhancer pairs, we first identified the pairs in which the 5′ and the 3′ enhancers respectively contained the motifs of interest or not. For each combination of motifs (200 × 200 = 40,000), we then computed the mean activity of the pairs that contained the motifs and normalized it to the activity of pairs that did not contain them. We also used this approach to systematically assess the impact of motifs on developmental enhancer super-additivity, using the mean residuals of the fitted multiplicative model with interaction term (see previous section), which reflect the difference between predicted multiplicative versus observed values.

To unbiasedly assess whether DNA binding motifs might be predictive of the activity of an enhancer pair, we trained a LASSO (using the glmnet R package, v4.1-4) regression to predict the activity of enhancer pairs using only motif counts (120 PWMs listed in Supplementary Data 8) and no other information. A similar model was used to predict multiplicative model's residuals (see previous section), to assess whether specific combinations of motifs might boost or dampen super-additive interactions between developmental enhancer pairs. For both approaches, we used the glmnet R package[34,35] (version 4.1-4). The cv.glmnet (alpha = 1, lambda = 10^seq(2, −3, by = −0.1), standardize = TRUE, nfolds = 5) function was used to infer the best lambda (bl) value to be passed to the glmnet (alpha = 1, lambda = bl, standardize = TRUE) to function and fit the model. Each model was evaluated using 9-fold cross-validation, ensuring non-overlapping held-out test sets.

## Over-representation of housekeeping/developmental among highly transcribed genes

Publicly available PRO-Seq data from *Drosophila* S2 cells were retrieved from GSE184187[22]. Only active genes with mean read counts higher than 1 at CAGE-TSS (and up to 150 bp downstream) were considered. We then used one-tailed Fisher's exact tests (alternative = "greater") to assess whether housekeeping or developmental genes (as defined in ref. 22) were over-represented among the top 10% of genes with the highest rates of transcription at their TSS.

## IDR content analysis

We used publicly available data reporting the activity of 812 *Drosophila* factors in 24 different enhancer contexts in S2 cells[25] to identify TFs and COFs that were sufficient to activate either the DSCP (developmental) or the RpS12 (housekeeping) core promoter, respectively (log2 fold change > log2(1.5)). Then, TFs/COFs that are expressed in S2 cells[1] (rpkm > 0.1) were classified based on their preference for either the housekeeping (RpS12 log2 fold change > DSCP log2 fold change) or the developmental (RpS12 log2 fold change < DSCP log2 fold change) core promoter[1]. This approach identified 74 and 52 TFs/COFs showing a preference for housekeeping and developmental contexts, respectively. For each of them, the fraction of predicted Intrinsically Disordered Regions (IDRs) and their length were retrieved from the MobiDB database[36].

## Luciferase assays

The promoter of the pGL3 Luciferase Reporter Vector (Promega) was replaced with the DSCP core promoter[4], and the resulting construct was used to validate pSTARR-seq measurements. A set of control-control, enhancer-control, control-enhancer and enhancer-enhancer pairs were amplified and flanked with overhangs compatible with Gibson assembly® (5′ overhang = ATTTCTCTATCGATAGGTAC. 3′ overhang = GTACCGAGCTCTTACGCGTC). Resulting sequences were cloned upstream of the core promoter (using KpnI restriction site) via Gibson assembly® (New England BioLabs, #E2611S). Each construct was verified using Sanger sequencing and luciferase assays were performed as previously described[1]. For each replicate, $6 \times 10^5$ S2 cells were transfected with a mix consisting of 100 ng of luciferase plasmid and 5 ng of Renilla plasmid per well using FuGENE® HD (Promega, #E2311). At least three biological replicates were measured per construct, each consisting of at least 3 technical replicates. Then, activities were normalized using control/control pairs. For homotypic enhancer pairs (Supplementary Fig. 2i), 5′ and 3′ individual activities were inferred from enhancer/control and control/enhancer pairs, respectively, and used to compute predicted additive outcomes, as for STARR-seq (see above).

## Statistics and reproducibility statement

Complying with the guidelines referenced in ref. 16, STARR-seq assays were performed in at least two biological replicates, meaning that independent transfections and downstream processing were performed on different days. Pearson Correlation Coefficients between replicates are reported in Supplementary Data 14. Luciferase assays were performed in at least three independent biological replicates, and the standard deviation between them was systematically shown on corresponding plots. No statistical method was used to predetermine sample size. No data were excluded from the analyses. The experiments were not randomized. The investigators were not blinded to allocation during experiments and outcome assessment.

## Reporting summary

Further information on research design is available in the Nature Portfolio Reporting Summary linked to this article.

## Data availability

Raw sequencing data and processed files generated for this study were deposited to GEO (GSE245033). Publicly available STARR-seq data from *Drosophila* S2 and OSC cells were obtained from GSE183939[17], GSE47691[19], GSE57876[4]. Publicly available DHS and RNA-Seq data from *Drosophila* S2 cells were obtained from GSE40739[1]. RNA-Seq data from OSC cells and ecdysone-treated S2 cells were obtained from GSE40739[1] and GSE47691[19], respectively. Publicly available PRO-Seq data was obtained from GSE184187[22]. Source data are provided with this paper.

## Code availability

All custom scripts to analyze the data and plot the figures are publicly available on GitHub: https://github.com/vloubiere/git_peSTARRSeq[37].

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

## Acknowledgements

We thank the IMP/IMBA/GMI core facilities for support and members of the Stark group and Yoav Voichek for feedback and discussion. Next-generation sequencing was done at the Vienna Biocenter Core Facilities GmbH (VBCF) Next-Generation Sequencing Unit. We thank Franziska K. Lorbeer for helping with cell culture. V.L. was supported by HFSP (LT000926/2020) and EMBO (790-2019) postdoctoral fellowships. Research in the Stark group is supported by Boehringer Ingelheim GmbH, the Austrian Research Promotion Agency (FFG, FO999902549) and the Austrian Science Fund (FWF, 10.55776/P29613, 10.55776/P33157, 10.55776/P36971 and 10.55776/PAT3564423). For the purpose of Open Access, the authors have applied a CC BY public copyright license to any Author Accepted Manuscript (AAM) version arising from this submission.

## Author contributions

V.L. and A.S. designed the experiments. V.L. developed fusion PCR protocols, generated enhancer-enhancer pairs libraries, and performed STARR-seq experiments together with M.P. V.L. performed bioinformatic analyses with the help of B.P.A. V.L. and A.S. wrote the manuscript, which was read and critically assessed by all authors.

## Competing interests

The authors declare no competing interests.
