## [Peer Review File · Nature Communications]

Developmental and housekeeping transcriptional programs display distinct modes of enhancer-enhancer cooperativity in *Drosophila*Reviewers' Comments:

Reviewer #1:

Remarks to the Author:

This is a concise and powerful study demonstrating a fascinating gene regulatory phenomenon in *Drosophila*: that developmental enhancers act multiplicatively, whereas housekeeping enhancers act additively. The authors design a new ‘enhancer x enhancer’ STARR-seq assay that measures the enhancer activity of pairwise combinations of 249-bp enhancer sequences. They measure a 1000x1000 matrix with a developmental core promoter, and smaller matrices with both a developmental and housekeeping gene core promoter. They use an appropriately simple modeling framework to show that the data for the dCP are fit very well by a multiplicative combination of the individual enhancer activities, whereas the hkCP data are fit very well by an additive combination of individual enhancer activities. The magnitude of the difference is striking. The study finds that there is some evidence of saturation at the high end of expression, and that there are no clear TF motifs that can explain residuals from the model. Finally, the study notes a difference in the ‘IDR fraction’ of TFs that prefer to activate housekeeping versus developmental core promoters, and propose this as a possible explanation.

Overall the study is exciting, elegant, and technically robust. The topic of whether enhancers act additively, super-additively, or sub-additively is one that is of great interest, and the finding that different types of enhancers combine differently is very interesting. I have only a small number of suggestions for improving the study.

1. Vocabulary and terminology. As the authors are aware, judging from their writing, the language around this topic is often confusing and confused in various studies. In general, the manuscript uses a good definition for studies of transcriptional regulation— which is where additive means that effect of X and Y together is the effect of X + effect of Y, in linear gene expression space. Some suggestions to consider:

1a. because other studies use different terminology, it would be great if the text could define terms with even more precision. E.g., “their combined transcriptional outcome mirrors the sum of their individual activities” could benefit from clarifying “in linear gene expression space”, and perhaps even a simple cartoon figure showing the effect.

1b. It might also be worth double checking whether the evidence presented in the studies cited as “super-additive”, “sub-additive” etc. indeed super-additive with respect to gene expression, or with respect something else.

1c. Is it correct that “additive” implies “independent” whereas “multiplicative” implies “not independent”? (this is implied in Line 139). This is potentially confusing because from a different frame of reference (e.g. fold-changes in gene expression), in the case of a purely multiplicative model, the effect of each enhancer does not depend on the other and could be viewed as “independent”. Similarly, I am not sure “synergistic” is fully appropriate. It might be easier to always

use the terms “multiplicative”, “super-additive”, and “additive”, and avoid “independent”, “synergistic”.

2. It could be helpful to provide some examples in the text to illustrate the difference between a multiplicative and additive model. For example, are there pairs of developmental and housekeeping enhancers that have approximately the same ‘activity’ when paired with the respective type of promoter, but that have clearly different effects when combined? A barplot showing some of these examples could be effective for readers to visualize the magnitude of the differences that can result from the additive vs multiplicative relationship

3. I have a suggestion for additional STARR-seq experiments to further understand the basis of the multiplicative versus additive models. I do not feel like these are necessary for publication, but could be interesting and straightforward next experiments that could refine the proposed model: Could you construct synthetic dev or housekeeping enhancers by adding motifs of TFs? E.g., how does activity of a single enhancer change if you have 0, 1, 2, 3, 4, 5, 6 copies of different TF motifs, either for TFs that preferentially activate developmental or housekeeping enhancers?

Minor comments:

- Line 55: “first step for a gene to exert its biological function” — should this be, “first step for a genetic variant to exert its biological function”? or “first step needed for a gene to exert its biological function”?
- Line 99: “know” should be “known”
- The rationale for defining “Strong” “Medium” “Weak” categories of enhancers is unclear from Fig 1b, since the groups are highly overlapping
- Fig 2: Are the axes limited to some maximum value? If so, for evaluating the trend where high-expression pairs are expressed less than the model predicts, it could be relevant to extend the axes farther.
- Line 107: Is it correct that the enhancer sequences are 249bp? Or are the oligos 249bp? It would be worth specifying the length of the enhancers.
- Supp Fig. 1c — The Ctl./Enh. pairs appear to have significantly different normalized luciferase activity than Enh./Ctl. pairs. Why would this be?
- Is “IDR fraction” equivalent to “IDR length”, or are there also systematic differences in the total amino acid length of TFs that prefer development vs housekeeping core promoters?
- Fig S2 — For clarity, could you specify which dataset this figure represents in the legend?

- I think that instead of “linear model”, it might be clearer to name it something else, e.g. “regression model” or “multiplicative model with interaction term”

Signed,
Jesse Engreitz

Reviewer #2:

Remarks to the Author:

In this manuscript, Loubiere et al. carried out a large-scale cloning and STARR-seq-based approach on *Drosophila* S2 cells to study enhancer-enhancer cooperativity. The main findings and conclusions are the developmental enhancer pairs activate target genes synergistically/multiplicatively, while the enhancer pairs of housekeeping genes cooperate additively. This is the first paper directly measure the enhancers' individual and combined activities, even though the STARR-seq method has been long established. The experimental scale in terms of the number of enhancer pairs is unprecedentedly large, even though the number of individual enhancers is around 1000. This study also includes around 1000 mutant and add-on enhancer variants to Twist/Trl and Dref motifs, to investigate the cooperative mode of TF/CP/enhancers. The overall approach and findings are novel and as such useful for the field. However, there are several important issues in the data QC and comparison that the authors should address to make all conclusions solid.

Main Points:

1. Line 51 “providing a rationale for strong and mild transcriptional effects of mutations within enhancer regions.” and Line 92-95

Care should be taken in over-interpreting the data drawn from the comparison between the wildtype and mutant enhancers only performed on Twist/Trl motifs in tens of enhancers. Larger number of motifs and enhancers are needed to draw such firm conclusion. Otherwise, alternative explanations should also be considered/discussed.

2. The impact of enhancer location at either 5' or 3' was not sufficiently compared. Fig. 1b only shows the comparison for individual activities of each enhancer at either 3' and 5'. The author should also compare the same pair of 2 enhances with just swapped 5' and 3' positions. Those enhancer pairs with dramatic difference due to 5'/3' location can be excluded from the following analyses. Or, the model-fitting analyses should be done separately on the enhancer pairs with different level of divergence.

In Fig. 1c, it is not clear if the difference between enhance/control and control/enhancer groups is statistically significant.

As also demonstrated in Fig. S2d-e, when there is a weakest enhancer at 5' or 3' enhancers, the

ability and dynamics of enhancers at the other location to increase the activity of the pair vary between 5' and 3'. However, this phenomenon was not mentioned in the main text, nor discussed with author's interpretation.

3. In the synergistic/multiplicative model of Developmental enhancer pairs, is there general difference among homotypic and heterotypic 5'/3' combinations?

4. Fig. 3b, Page 6 Line 191-203. It is not clear if all the enhancer pairs containing the motifs, Trl and Twist, for example, all associate with residuals or not. If not only with residuals, how they behave in the non-residual part. The comparative analyses in Fig. 3c, 3d should be done not only on residuals but also on the activity in the non-residual part.

To draw the conclusion in Line 199 "developmental enhancer synergy does not strongly rely on these two motifs.", the author should systematically compare both residuals and activities among WT, Mutant, and Added Motif of enhancer pairs. Also, the conclusions in Line 204-209, Page 7, are over-interpreted since the reporter experiment cannot fully mimic the regulatory effects of motif sequences and distances (less or more than 20kb apart) etc. as in the endogenous loci.

5. As the authors pointed out in this paper (Line 212-213) and previous publications, developmental- and housekeeping-type enhancers render different regulatory effects with developmental CP or housekeeping CP. Very nicely, the authors cloned a subset of 62 housekeeping enhancers, 53 developmental enhancers and 50 control sequences, individually and in pairwise, under either hkCP and dCP and tried to compare the consequent activities side-by-side. However, the only comparative result represented in Fig.4c was not sufficient enough to draw the conclusions (Line 237-243). Apart from the comparison of HK/HK and Dev/Dev enhancer pairs under Additive modes (Fig.4c), the other combinatory pairs including HK/Dev and Dev/HK, should be all compared under hkCP and dCP to both Additive and Synergistic models. As the authors have pointed out in the working model that the developmental enhancers combine multiplicatively until they eventually saturate the CP (Line 252), it would be essential to demonstrate the activity differences among different type of enhancer pairs under developmental and house-keeping CP.

6. The analysis on IDR percentage is very preliminary and the difference found (Fig. 4d) is far from sufficient to show any impact on the synergistic/multiplicative and additive mode of cooperation. This part should be removed from the main figure and discussed only as a perspective.

7. Line 293-297. From the results of this paper, the cooperation activities of housekeeping enhancer pairs appear to be generally lower than developmental enhancer pairs under their respective CP. Is it due to the varied characteristics e.g. initiation and/or saturation of hkCP and dCP? Based on the conclusion of this paper that HK/HK enhancers fit better to the additive model, what is the possible

force to boost the transcription of housekeeping genes generally higher than developmental genes?

8. The total number of wildtype enhancers (600 developmental enhancers and 100 housekeeping enhancers) is relatively small compared to over 50,000 developmental enhancers genome-wide (Kvon et al. Nature 2014). There are 2 main questions need to be addressed. 1). If these enhancers activate their gene targets synergistically or additively in their native genomic loci? 2). If the synergistic and additive modes of activation hold true at genome-wide scale? The authors should at least investigate the correlation of the 700 cloned enhancers and their possible pairs in the genomic loci to the endogenous expression level of their gene targets in S2 cells, to see if they fit into the same modes of cooperation achieved from STARR-seq data. In principle, this could be also investigated at genome-wide scale, using mathematic modeling or machine learning approach.

9. The authors design and cloned several Inducible enhancers and OSC-specific enhancers; however, no relevant results, findings, and discussions are presented. The same experiments should be done at least on subsets of relevant enhancers under inducible conditions (Ecdysone and Heatshock) on S2 cells, and on OSCs, to see if they fit the same synergistic/multiplicative or additive mode.

Minor points:

- 1) The limitation of method used in this work should be addressed.
- 2) Page 5, line 144, multiplicative model was more accurate for “88%”, rather than 90%, if written in a precise way.
- 3) Line 501-505, the list of selected motifs is not included in mentioned table/column.
- 4) Line 546, starts with a typo “l”.
- 5) Fig. 3c, 3d, "twist" should be "Twist". In the legend, “In situ” should be italic and “in pink” contains typo.

Reviewer #3:

Remarks to the Author:

Genes are often regulated by more than one enhancer, however how these enhancers interact in different biological contexts to coordinate gene expression is an open area of investigation. In this manuscript, Loubiere et al. screen a STARR-seq library of individual and paired enhancers for activity in Drosophila S2 cells to examine how developmental enhancers and housekeeping gene

enhancers interact to regulate gene expression. The authors' expertise in creating MPRA libraries is highlighted by a clever fusion PCR-based approach to create their paired enhancer STARR-Seq library. This library used a developmental core promoter and was composed of enhancers paired with inert controls (individual enhancers) and enhancer-enhancer pairs, which allowed the authors to compare reporter gene expression in both contexts. Through modeling of their STARR-Seq activity data, the authors find that most developmental enhancers interact in a synergistic manner, while most housekeeping enhancers interact additively in S2 cells. The authors conclude that the synergistic interaction between developmental enhancers is promiscuous in S2 cells, meaning that it is not dependent on specific transcription factor motifs, genomic location, or even promoter used. However, the authors do note that developmental enhancers are more enriched in binding sites for transcription factors with intrinsically disordered regions. This manuscript is generally scientifically sound, but I believe the following major and minor comments should be addressed to provide evidence for the authors' claims and to provide clarity for readers.

Major Comments/Concerns:

1. While S2 cells are a great starting point for examining this biological question, it is difficult to believe the authors generalized claims about interactions of developmental enhancers outside of a developmental context, where exposure to different activating and repressive TFs is varied across time and space. For example, in the authors cited reference 10, Bothma et al. state that the hunchback developmental enhancer behaves differently (additively vs sub-additively) depending on the cellular context (levels of Bicoid); this is somewhat captured in the Loubiere et al. STARR-Seq data showing CP saturation when a strong enhancer is used. Additionally, in citation 14, Lam et al. describe how two enhancers for the proopiomelanocortin gene in mice initially act synergistically in embryos and as development progresses to adulthood, these enhancers behave additively. Without some type of developmental biological variable used (different cell lines, synthetic signaling gradients, or a handful of examples using BAC transgene reporters in flies or other system), it is difficult to accept a generalized claim about developmental enhancers with data gathered only from a single cell line.

a. It would be nice to see, if the sequences are in the dataset, if enhancers that were previously examined by others behave as previously reported. This might add some weight to the authors claims about developmental enhancers.

2. In your library, how do you account for non-similar genomic spacing? The authors saw no difference between genomically close and distant linked enhancers using a 300bp spacer, but does spacing between the enhancers change the way they interact? The synthetically close spacing may perhaps be more similar for housekeeping enhancers (as noted in the text), but what about developmental enhancers? A test of a handful of enhancers that interact endogenously to examine the 300bp spacing used by the authors and comparing this to endogenous spacing would be a useful control.

3. The authors claim that TF motifs likely only play a very minor role, if any, in the synergistic behavior of developmental enhancers. They then go on to test this claim by examining TF motifs present in enhancer pairs with activity beyond what their model predicts (residuals). They mutate

Trl and Twist sites in active enhancers that contain them or by adding them into their library of enhancers (inert or active sequences). Lines 379 to 386 in the methods section raise some concerns about the experimental approach, but these could just be points of clarification.

a. It is stated that, “we mutated Twist, Trl and Dref motifs by replacing them with random stretches of nucleotides within a set of active enhancers that contained them.”

i. Were the random sequences consistent for each TF site replaced? It seems like multiple were initially used, but how many were actually selected in the end?

ii. Were the random sequences that were used as “mutated sites” tested for activity or checked to ensure that a new binding site was not created?

b. Lines 381 through 386 detail that many sequence iterations were used per mutation or site addition and those predicted to have the least impact on individual enhancer activity were selected for analysis.

i. This is very confusing for the reader as it seems the data that was chosen to be analyzed was biased towards the null hypothesis for these TFs having an impact on enhancer interaction. Please clarify or explain why this approach was done over using the median or some type of averaging.

4. The authors show a small glimpse that binding sites for IDR TFs could be driving developmental enhancers to behave synergistically compared to housekeeping enhancers. A simple experiment here is to add these sites into housekeeping enhancers and use qPCR or a Luciferase assay to see if it causes shifts to a more synergistic interaction. Wrapping the story up with a mechanism would bolster the impact of this manuscript.

Minor Comments/Clarifications:

1. While the authors claim that specific TF binding sites largely don't impact the behavior of enhancer interactions as a whole, could something like binding site affinity (similar to levels of Bicoid in Bothma et al., 2015) impact behavior?

2. Lines 200 through 203:

“Consistently, a LASSO regression using motif counts as input was able to predict the overall activity of enhancer pairs ($R^2= 0.37$) but not the residuals ($R^2= 0.08$, Supplementary Fig. 3a-b), confirming the association between TF motifs and enhancer activities and suggesting that residuals or synergies do not rely on specific DNA binding motifs.”

i. With the low R^2 value, I don't agree with the statement the LASSO was able to predict enhancer pair activity, thus I don't believe the authors should use this to suggest anything about TF motifs and enhancer activities or their impact on residuals.

3. Line 352 “flanked by PCR primers” should be “flanked by PCR primer binding sites”.

4. Line 438 “indexe” should be “index”.

5. Line 545 “tree” should be “three”.

6. Figure 2D legend: “Boxplot showing, for all enhancer pairs, expected (right) and observed activity values (left) using the three different models.”

a. I believe left and right are meant to be swapped.

7. Supplementary Figure 3D is mislabeled as C in the legend and is missing statistical test information.

List of main changes:

We thank all three reviewers for their constructive and helpful comments that allowed us to extend the scope of our study and improve the clarity of the manuscript. In the revised version of the manuscript, we have thoroughly addressed all the pending questions and comments from the three reviewers.

We implemented a new analysis to show that the enhancer location either at the 5' or the 3' end had no substantial impact on activity (new Fig. 1e), and performed a new STARR-Seq assay with a 2kb spacer to show that increased genomic distance did not prevent super-additive interactions between developmental enhancers (new Supp. Fig. 2f-h).

We also performed two new STARR-Seq assays in hormone-treated S2 cells and in *Drosophila* Ovarian Somatic Cells (OSCs). Importantly, we found hormone-inducible and OSC-specific enhancers to also be super-additive (new Fig. 2f-g). We also sequenced STARR-seq libraries for the motif-mutant enhancers more deeply (Fig. 3c-d) to increase overall statistical power and the number of pairs for which individual activities could be robustly measured. We also revised the corresponding representation to more clearly show that mutating Trl or Twist motifs (1) does not abolish super-additivity but that (2) Trl and Twist motifs slightly boost super-additive outcomes.

Regarding motif cooperativity within individual enhancers, we introduced a new analysis to show that, similar to developmental motifs, Dref housekeeping motifs combine multiplicatively within individual housekeeping enhancers (Sup. Fig. 3i-k), contrasting with the additive behavior that characterizes housekeeping enhancer pairs.

Overall, these new results confirm our findings that super-additivity is a promiscuous feature of developmental enhancers in *Drosophila*, and does not seem to rely on strict motif syntax rules.

REVIEWER COMMENTS

Reviewer #1 (Remarks to the Author):

This is a concise and powerful study demonstrating a fascinating gene regulatory phenomenon in *Drosophila*: that developmental enhancers act multiplicatively, whereas housekeeping enhancers act additively. The authors design a new ‘enhancer x enhancer’ STARR-seq assay that measures the enhancer activity of pairwise combinations of 249-bp enhancer sequences. They measure a 1000x1000 matrix with a developmental core promoter, and smaller matrices with both a developmental and housekeeping gene core promoter. They use an appropriately simple modeling framework to show that the data for the dCP are fit very well by a multiplicative combination of the individual enhancer activities, whereas the hkCP data are fit very well by an additive combination of individual enhancer activities. The magnitude of the difference is striking. The study finds that there is some evidence of saturation at the high end of expression, and that there are no clear TF motifs that can explain residuals from the model. Finally, the study notes a difference in the ‘IDR fraction’ of TFs that prefer to activate housekeeping versus developmental core promoters, and propose this as a possible explanation.

Overall the study is exciting, elegant, and technically robust. The topic of whether enhancers act additively, super-additively, or sub-additively is one that is of great interest, and the finding that different types of enhancers combine differently is very interesting. I have only a small number of suggestions for improving the study.

We thank the reviewer for highlighting the interest of the topic and the general appreciation of our work. We hope that the following point-by-point response will help clarify pending questions.

1. Vocabulary and terminology. As the authors are aware, judging from their writing, the language around this topic is often confusing and confused in various studies. In general, the manuscript uses a good definition for studies of transcriptional regulation—which is where additive means that effect of X and Y together is the effect of X + effect of Y, in linear gene expression space. Some suggestions to consider:

1a. because other studies use different terminology, it would be great if the text could define terms with even more precision. E.g., “their combined transcriptional outcome mirrors the sum of their individual activities” could benefit from clarifying “in linear gene expression space”, and perhaps even a simple cartoon figure showing the effect.

Thanks for this proposition. We agree and now modified line 72 to state that: “Early studies suggested that enhancers are additive [7–9], meaning that their combined transcriptional outcome mirrors the sum of their individual activities in linear gene expression space, i.e. the number of resulting RNA molecules add up”. We also added a cartoon in Fig. 2a to illustrate simple additive and multiplicative outcomes (referenced in line 151 of the revised manuscript).

1b. It might also be worth double checking whether the evidence presented in the studies cited as “super-additive”, “sub-additive” etc. indeed super-additive with respect to gene expression, or with respect something else.

We thank the reviewer for highlighting that some of the original references were not super-additive with respect to gene expression but to something else. We replaced the reference to Chatterjee et al. (2016) by a recent, tightly controlled dissection of the alpha-globin locus pointing at synergistic transcriptional effects (Blayney et al., reference 11, line 74). We also removed the reference to Bahr et al. (2018, line 72), in which additive outcomes were only briefly mentioned.

1c. Is it correct that “additive” implies “independent” whereas “multiplicative” implies “not independent”? (this is implied in Line 139). This is potentially confusing because from a different frame of reference (e.g. fold-changes in gene expression), in the case of a purely multiplicative model, the effect of each enhancer does not depend on the other and could be viewed as “independent”. Similarly, I am not sure “synergistic” is fully appropriate. It might be easier to always use the terms

“multiplicative”, “super-additive”, and “additive”, and avoid “independent”, “synergistic”. We agree that it is easier to always use the terms “multiplicative”, “super-additive” and “additive”, which derive directly from our observations, and we revised the manuscript accordingly.

We also agree that, in the case of a purely multiplicative model, the effect of each enhancer could be viewed as independent, and we removed our previous mechanistic statement (line 148). We now write: “An additive model posits that the combined enhancer activity is the sum of the individual enhancer activities, i.e. that the numbers of RNA molecules produced add up. Conversely, the multiplicative model posits that the enhancer activities, and thus the number of RNAs, behave multiplicatively (Fig. 2a)”.

2. It could be helpful to provide some examples in the text to illustrate the difference between a multiplicative and additive model. For example, are there pairs of developmental and housekeeping enhancers that have approximately the same ‘activity’ when paired with the respective type of promoter, but that have clearly different effects when combined? A barplot showing some of these examples could be effective for readers to visualize the magnitude of the differences that can result from the additive vs multiplicative relationship

We agree and now added a panel showing examples of 20 housekeeping and developmental enhancer pairs which were selected to have similar 5’ and 3’ activities on their own. The measured combined activity (grey) compared to the expected additive or multiplicative activities (striped) reveal additivity for the housekeeping enhancers and multiplicativity for the developmental enhancers (new Fig. 4c):

Figure 4c: Selected housekeeping (left) and developmental (right) enhancer pairs with comparable 5’ and 3’ individual activities, either with a housekeeping (hkCP, in red) or a developmental (dCP, in green) Core Promoter. For each pair, individual and combined measured activities are shown (solid grey bars) and compared to predicted activities (striped bars) using either the additive (Pred. add.) or the fitted multiplicative (Pred. fit. mult.) model. Bar heights correspond to the mean activity values and whiskers to the standard deviations.

3. I have a suggestion for additional STARR-seq experiments to further understand the basis of the multiplicative versus additive models. I do not feel like these are necessary for publication, but could be interesting and straightforward next experiments that could refine the proposed model: Could you construct synthetic dev or housekeeping enhancers by adding motifs of TFs? E.g., how does activity of a single enhancer change if you have 0, 1, 2, 3, 4, 5, 6 copies of different TF motifs, either for TFs that preferentially activate developmental or housekeeping enhancers?

We agree that it is interesting to assess how the activity of a single enhancer changes depending on the number of TF motifs. We tackled this question using STARR-seq data from our previous work in S2 cells, which tested 249bp-long enhancers with developmental and housekeeping promoters (see ref. 17; <https://doi.org/10.1038/s41588-022-01048-5>). To assess how the activity of enhancer scales with the presence of developmental/housekeeping motifs, we stratified the enhancers based on the number and types of TF motifs they contain (new Supplementary Fig 3i-k):

Supplementary Figure 3i-k: - Enhancer activity of candidate sequence in S2 cells as a function of the number of Dref motifs they contain, using either a housekeeping (hkCP, in red) or a developmental (dCP, in green) Core Promoter. j- Same as h, but looking at AP-1 developmental motifs. k- Enhancer activity of candidate sequences in S2 cells depending on whether they contain at least one instance of each of the motif listed on the x axis, using either a housekeeping (hkCP, in red) or a developmental (dCP, in green) Core Promoter. “None” corresponds to the sequences that did not contain any of these motifs.

We now report this observation and its implications in the revised manuscript (line 320): “Interestingly, for the different TF motifs within a single enhancer, we confirmed the TF motifs’ promoter-selectivity and their multiplicativity – sometimes referred to as motif cooperativity or synergy¹⁷ – for both developmental and housekeeping enhancers (Supplementary Fig. 3i-k). These results indicate that the modes of cooperativity between different housekeeping and developmental enhancers do not reflect the cooperativity of cognate TFs within individual enhancers”.

Minor comments:

- Line 55: “first step for a gene to exert its biological function” — should this be, “first step for a genetic variant to exert its biological function”? or “first step needed for a gene to exert its biological function”?

We corrected the sentence to “first step needed for a gene to exert its biological function”.

- Line 99: “know” should be “known”

Fixed (now at line 103).

- The rationale for defining “Strong” “Medium” Weak” categories of enhancers is unclear from Fig 1b, since the groups are highly overlapping

These categories were meant as an additional reference to illustrate the concordance of measuring the enhancers’ activities in a standard STARR-seq setup (from our previous work; see ref. 17) compared to the activities in the 5’ and 3’ positions of the STARR-seq setup used here. We now replaced these categories with the rank of these 953 individual sequences:

Figure 1b: Correlation between 3’ (x axis) and 5’ (y axis) individual activities of 953 candidate sequences. The dotted line represents the identity line (y=x) and the Pearson’s correlation coefficient is shown on the top left (r). The color code displays sequences’ activity rank inferred from a previously published STARR-seq dataset (ref. 17).

- Fig 2: Are the axes limited to some maximum value? If so, for evaluating the trend where high-expression pairs are expressed less than the model predicts, it could be relevant to extend the axes farther.

In the previous version, axes were clipped using 0.001 and 0.999 quantiles (of note, R squared values were always computed on the full dataset). We now removed this clipping, except for the x axis of the additive model, which was cut at -5 to prevent 5 low outliers from compressing the visible range. Of note, the x axes of Supplementary Fig. 2c-e dedicated to CP saturation are not clipped.

• Line 107: Is it correct that the enhancer sequences are 249bp? Or are the oligos 249bp? It would be worth specifying the length of the enhancers.

We now revised line 111 to be more precise, stating that: “we designed a pool of 300-bp oligos (249-bp candidate sequences flanked by PCR primer binding sites) [...]”.

• Supp Fig. 1c — The Ctl./Enh. pairs appear to have significantly different normalized luciferase activity than Enh./Ctl. pairs. Why would this be?

Ctl./Enh. pairs and Enh./Ctl. tested in luciferase assays were randomly selected and do not contain the same enhancers (in other terms, they do not correspond to A/B and reciprocal B/A pairs). As such, the differences observed with luciferase assays were also present in STARR-seq, albeit less pronounced:

Figure to reviewer: comparison of normalized luciferase vs STARR-Seq fold changes
Quantification of normalized luciferase (left) versus STARR-Seq fold changes for enhancer/control (Enh./Ctl., in blue) and control/enhancer (Ctl./Enh., in purple).

The remaining differences might be due to the different constructs used for these two assays: as we shifted the enhancer-spacer-enhancer block to a position upstream of the promoter, the 3' location is now closer to the CP compared to STARR-seq. To clarify this point, we now added cartoons of these two constructs next to the corresponding axes:

Supplementary Fig. 1c: Correlation between STARR-seq (x axis) and luciferase (y axis) measurements for a set of control-control random sequences (Ctl./Ctl., in grey), one control sequence paired with a candidate sequence either in the 5' (Enh./Ctl., in blue) or the 3' (Ctl./Enh., in purple) location, or two enhancer sequences (Enh./Enh., in green). Schematic views of the reporter constructs used for luciferase (top) or STARR-seq assay (right) are shown, as well as the Pearson's correlation coefficient (r , on the top left of the scatterplot).

We now also report that “the activities of reciprocal pairs (A/B versus B/A) were overall similar and highly correlated ($r= 0.80$)”, see line 139 new Fig. 1e:

Fig. 1e: Correlation between candidate sequence pairs (A/B, x axis) and the reciprocal combinations (B/A, y axis).

- Is “IDR fraction” equivalent to “IDR length”, or are there also systematic differences in the total amino acid length of TFs that prefer development vs housekeeping core promoters?

Thanks for this interesting question. The absolute length of IDRs (in aa) is longer in developmental TFs and COFs and the relative fraction of the proteins’ sequences that are IDRs is also higher, i.e. developmental TFs and COFs have more of their protein sequences dedicated to IDRs. We now show both comparisons as Supplementary Fig. 3l, following a request by reviewer 2 (point 6):

Supplementary Fig. 3l: Absolute length (in aa, left) or fraction (right) of Intrinsically Disordered Regions (IDRs) within TF and COFs proteins that preferentially activate Housekeeping (Hk., orange) or Developmental (Dev., green) promoters according to24. Two-sided Wilcoxon test P-values are shown.

- Fig S2 — For clarity, could you specify which dataset this figure represents in the legend?

Thanks for pointing this out. To clarify this point, we now titled supplementary figure 2 “strong developmental enhancer pairs saturate the CP” and reworded the captions of panel a and b to now read: “a- R-squared (R2) values for the 3 different models (bottom) in the developmental setup (developmental enhancer pairs downstream of a developmental CP). Higher R2 means better fit. b- Fraction of developmental enhancer pairs (in which both candidate sequences are active) for which the additive (in white), the multiplicative (in blue) or the fitted multiplicative model with interaction term (in pink) were the most accurate at predicting observed activities.”.

- I think that instead of “linear model”, it might be clearer to name it something else, e.g. “regression model” or “multiplicative model with interaction term”

We agree that “multiplicative model with interaction term” is clearer and revised the text accordingly (starting from line 159).

Signed,
Jesse Engreitz

Reviewer #2 (Remarks to the Author):

In this manuscript, Loubiere et al. carried out a large-scale cloning and STARR-seq-based approach on Drosophila S2 cells to study enhancer-enhancer cooperativity. The main findings and conclusions are the developmental enhancer pairs activate target genes synergistically/multiplicatively, while the enhancer pairs of housekeeping genes cooperate additively. This is the first paper directly measure the enhancers' individual and combined activities, even though the STARR-seq method has been long established. The experimental scale in terms of the number of enhancer pairs is unprecedentedly large, even though the number of individual enhancers is around 1000. This study also includes around 1000 mutant and add-on enhancer variants to Twist/Trl and Dref motifs, to investigate the cooperative mode of TF/CP/enhancers. The overall approach and findings are novel and as such useful for the field.

However, there are several important issues in the data QC and comparison that the authors should address to make all conclusions solid.

We thank the reviewers for their appreciation of our work and their helpful feedback. Please find below our point-by-point response to the different issues that were raised.

Main Points:

1. Line 51 “providing a rationale for strong and mild transcriptional effects of mutations within enhancer regions.” and Line 92-95

Care should be taken in over-interpreting the data drawn from the comparison between the wildtype and mutant enhancers only performed on Twist/Trl motifs in tens of enhancers.

Larger number of motifs and enhancers are needed to draw such firm conclusion. Otherwise, alternative explanations should also be considered/discussed.

We thank the reviewer for pointing out that the scope of these statements was unclear. We were referring to the stronger transcriptional impact of non-coding mutations affecting super-additive developmental enhancers compared to additive housekeeping enhancers. We agree with the need for clarification and rephrased this sentence to “These results have important implications for our understanding of gene-regulation in complex multi-enhancer developmental loci and genomically clustered housekeeping genes, providing a rationale to interpret the transcriptional impact of non-coding mutations at different loci” (line 52).

2. The impact of enhancer location at either 5' or 3' was not sufficiently compared. Fig. 1b only shows the comparison for individual activities of each enhancer at either 3' and 5'. The author should also compare the same pair of 2 enhancers with just swapped 5' and 3' positions. Those enhancer pairs with dramatic difference due to 5'/3' location can be excluded from the following analyses. Or, the model-fitting analyses should be done separately on the enhancer pairs with different level of divergence.

In Fig. 1c, it is not clear if the difference between enhancer/control and control/enhancer groups is statistically significant.

Thanks for these remarks. We now assessed the difference between Enh./Ctl. and Ctl./Enh. pairs (Fig 1c) and found that it is indeed statistically significant:

Fig. 1c: Quantification of the activity of pairs consisting of two random control sequences (Ctl./Ctl., in grey), one control sequence paired with a candidate sequence either in the 5' (Enh./Ctl., in blue) or the 3' (Ctl./Enh., in purple) location, or two enhancer sequences (Enh./Enh., in green). Two-sided Wilcoxon test P-values are shown.

However, the difference between these two groups is very small, suggesting that swapping the location of the enhancers had no substantial impact on activity. Consistently, the comparison of the same pair of 2 enhancers with swapped 5' and 3' positions proposed by the reviewer indicated a high concordance ($r = 0.80$), which we now show in new Fig. 1e (referenced line 139):

Fig. 1e: Correlation between candidate sequence pairs (A/B, x axis) and the reciprocal combinations (B/A, y axis).

As also demonstrated in Fig. S2d-e, when there is a weakest enhancer at 5' or 3' enhancers, the ability and dynamics of enhancers at the other location to increase the activity of the pair vary between 5' and 3'. However, this phenomenon was not mentioned in the main text, nor discussed with author's interpretation.

Regarding Supplementary Fig. 2 d-e, we initially compared the activity of pairs containing either the single weakest 5' enhancer (paired with 183 different 3' enhancers) or the single 3' weakest enhancer (paired with 124 different 5' enhancers). Because the number, identity and the individual activities of the paired enhancers were different for the two analyses (old Fig. S2d-e), the trends observed for 5' and 3' locations could not be compared side-by-side.

We agree that this might have been unclear and that a direct comparison would be valuable. We therefore opted for a more robust approach, where we went from the single weakest enhancer to a group of more than a hundred weak enhancers (\log_2 individual activity between 1 and 1.5), either in the 5' or the 3' location. This way, we could now compute the mean activity of pairs containing a weak 5' enhancer across 818 different 3' enhancers, and the mean activity of pairs containing a weak 3' enhancer across 825 5' enhancers, allowing direct comparison between the two:

Supplementary Fig. d- Individual activity of 818 enhancers in the 3' position (x-axis) versus their average combined activities (y axis) when paired with a weak (individual activity between 1 and 1.5; $n = 145$; left scatterplot) or a strong (individual activity $>$ strongest individual activity $- 1$; $n = 8$; right scatterplot) 5' enhancer. While weak 5' enhancers can be strongly boosted by the 3' enhancer, pairs containing strong 5' enhancers plateau around eight. e- Same as d but considering 825 enhancers located in the 5' position, paired with weak ($n = 110$, left scatterplot) or strong ($n = 7$, right scatterplot) 3' enhancers.

With this approach, we didn't see any clear difference between the two locations, which we now summarize (line 173): "Consistently, the activities of the strongest 5' or 3' enhancers can hardly be increased by the addition of a second enhancer (Supplementary Fig. 2d-e)."

3. In the synergistic/multiplicative model of Developmental enhancer pairs, is there general difference among homotypic and heterotypic 5'/3' combinations?

Due to the PCR-based processing of STARR-seq libraries, homotypic pairs cannot be assessed reliably (presumably due to template switching) and we thus excluded such pairs (see Methods, line 609). To assess the reviewer's question, we now performed luciferase assay for 10 different homotypic pairs as well as related control pairs (Control/Control, Enhancer/Control, Control/Enhancer and Enhancer/Enhancer), which behaved super-additively as expected (new Supplementary Fig. 2i, referenced line 180):

Supplementary Fig. 2i: Predicted additive (x axis) versus observed (y axis) combined activities inferred using luciferase assays for 10 different homotypic enhancer pairs, in which 5' and 3' individual enhancers are the same. The identity line ($y=x$) is shown, corresponding to the additive model.

We further assessed enhancer pairs in which both enhancers contain similar sets of motifs (which could be seen as homotypic) versus pairs in which the two enhancers contain different sets of motifs and found no systematic difference (new Supplementary Fig. 3a):

Supplementary Fig. 3a: Mean residuals difference (inferred using the fitted multiplicative model with interaction term) associated to homotypic (left) versus heterotypic (right) combinations of TF motifs (in which 5' and 3' enhancers either contain

instance(s) of the same motif or of two different motifs, respectively). Twist/ Twist pairs containing at least one Twist motif in their 5' and 3' enhancers globally show higher residuals compared to AP-1/AP-1 pairs. Two-sided Wilcoxon test P-value is shown.

4. Fig. 3b, Page 6 Line 191-203. It is not clear if all the enhancer pairs containing the motifs, Trl and Twist, for example, all associate with residuals or not. If not only with residuals, how they behave in the non-residual part. The comparative analyses in Fig. 3c, 3d should be done not only on residuals but also on the activity in the non-residual part.

To draw the conclusion in Line 199 "developmental enhancer synergy does not strongly rely on these two motifs.", the author should systematically compare both residuals and activities among WT, Mutant, and Added Motif of enhancer pairs. Also, the conclusions in Line 204-209, Page 7, are over-interpreted since the reporter experiment cannot fully mimic the regulatory effects of motif sequences and distances (less or more than 20kb apart) etc. as in the endogenous loci.

We thank the reviewer for this feedback, which motivated us to critically revisit this entire analysis. We first sequenced the mutant STARR-seq library to increase the number of enhancer pairs for which individual activities could be robustly inferred and thus improve the overall statistical power. Following the reviewer's suggestion, we then presented the dataset more exhaustively to show both the activity and the residuals of WT and mutated pairs side-by-side (Fig.3 c-d):

Figure 3c-d: Impact of mutating (c, in pink) or adding (d, in pink) Trl (left) or Twist motifs (right) on the predicted additive (x axis) versus observed combined activities (y axis) of developmental enhancer pairs. As a reference, corresponding wild-type (WT) developmental enhancer pairs are shown in grey. Dotted lines depict the identity line (where observed combined activities equal expected additive outcomes) and plain lines represent the fitted multiplicative model. For each condition, the residuals of WT versus mutant pairs were quantified (see boxplots on the right) using either the additive or the fitted multiplicative model (Fit. mult., see x axis) and compared using paired, two-sided Wilcoxon tests.

As suspected by the reviewer, not all WT enhancer pairs containing Twist or Trl motifs show positive residuals (the grey dots distribute roughly indiscriminately around the fitted multiplicative model in Fig. 3c). Moreover, mutating or pasting Trl or Twist motifs had little impact on super-additivity, as most variant pairs (in pink) remained super-additive (see pink dots and dotted lines in Fig. 3c and 3d, respectively). However, our reanalysis showed that Trl and Twist motifs slightly boost super-additivity: mutating Trl/Twist motifs moderately, yet statistically significantly decreased residuals and pasting the motifs increased the residuals relative to the additive and the fitted multiplicative model (see boxplots). We summarized this finding (line 220): "although their motifs are dispensable for super-additivity, Trl and Twist TFs slightly boost such interactions in S2 cells".

We also thank the reviewer for pointing out that the comparison of endogenously close (<20kb) versus more distant enhancers (>20kb) as an additional control had not been explained sufficiently clearly. We

do not want to claim that we measure cooperativity between enhancers at such distances and only selected these two groups of enhancers as control groups; we now rephrased this part (line 233): “To test whether enhancer pairs that are found in the same locus ($\leq 20\text{kb}$ apart) and could cooperate *in situ* might have evolved to enhance their super-additivity (*via* means that would not be captured by classical motif analyses), we compared such pairs to a control set of more distant enhancers ($>20\text{kb}$ in situ), and found no substantial difference in our system (Fig. 3e)”.

5. As the authors pointed out in this paper (Line 212-213) and previous publications, developmental- and housekeeping-type enhancers render different regulatory effects with developmental CP or housekeeping CP. Very nicely, the authors cloned a subset of 62 housekeeping enhancers, 53 developmental enhancers and 50 control sequences, individually and in pairwise, under either hkCP and dCP and tried to compare the consequent activities side-by-side. However, the only comparative result represented in Fig.4c was not sufficient enough to draw the conclusions (Line 237-243). Apart from the comparison of HK/HK and Dev/Dev enhancer pairs under Additive modes (Fig.4c), the other combinatory pairs including HK/Dev and Dev/HK, should be all compared under hkCP and dCP to both Additive and Synergistic models.

As the authors have pointed out in the working model that the developmental enhancers combine multiplicatively until they eventually saturate the CP (Line 252), it would be essential to demonstrate the activity differences among different type of enhancer pairs under developmental and house-keeping CP.

Following the reviewer’s suggestion, we compared the dev/hk and hk/dev enhancer pairs downstream of developmental and housekeeping CP (all combinations), and found that developmental enhancers remain super-additive in the context of a housekeeping enhancer/CP pair (see line 307 and new Supplementary Fig. 3h):

Supplementary Fig. 3h: Expected additive and observed activities of housekeeping/developmental enhancer pairs (x axis) using either a housekeeping (hkCP, in red) or a developmental (dCP, in green) Core Promoter. Two-sided Wilcoxon test P-values are shown.

6. The analysis on IDR percentage is very preliminary and the difference found (Fig. 4d) is far from sufficient to show any impact on the synergistic/multiplicative and additive mode of cooperation. This part should be removed from the main figure and discussed only as a perspective.

While the differences in IDR percentage and in absolute IDR lengths (see comment by reviewer 1) are both significant, we follow the reviewer’s suggestion and now moved this panel to Supplementary Fig. 3l. We also revised our discussion to state that “further studies would be needed to tackle how additive versus super-additive behaviors are encoded at the protein level, and whether other chromatin-related features might further constrain enhancer cooperativity *in situ*[12], and how motif binding affinity might influence these interactions[8]” (line 334).

7. Line 293-297. From the results of this paper, the cooperation activities of housekeeping enhancer pairs appear to be generally lower than developmental enhancer pairs under their respective CP. Is it due to the varied characteristics e.g. initiation and/or saturation of hkCP and dCP? Based on the conclusion of this paper that HK/HK enhancers fit better to the additive model, what is the possible force to boost the transcription of housekeeping genes generally higher than developmental genes?

Thanks for raising this point. The hkCP does not reach saturation in our assay and the differences we

see are unlikely to result from CP saturation because we observed no plateau even for the highest activities:

Figure 4a: Scatterplots showing predicted activities (x axis) based on an additive (left) or a multiplicative model (right) versus observed activities (y axis) using the RpS12 housekeeping CP. Corresponding R -squared (R^2) values are shown (top left) and enhancer pairs in which both candidate sequences are active are highlighted using density lines (in orange). Identity lines are shown using dotted lines ($x=y$).

We agree that housekeeping genes have high steady-state transcript levels as for example detected by mRNA-seq. Based on the literature, this seems to largely stem from increased mRNA stability (e.g. Faucillion et al., NAR 2022; doi: [10.1093/nar/gkac208](https://doi.org/10.1093/nar/gkac208)), while developmental genes show the highest transcription rates (e.g. Lubliner et al., NAR 2013; Arnold et al., Nature Biotech 2017). Following the reviewer’s suggestion, we now compared transcription rates in S2 cells based on nascent transcript sequencing with PRO-seq. This confirmed that among the most highly transcribed genes, developmental genes are highly enriched (new supplementary Fig. 3g):

Supplementary Fig 3g: Odds ratio of developmental and housekeeping genes among the top 10% expressed genes in S2 cells, defined using PRO-Seq data. One-tailed Fisher’s exact test P -values are shown.

We now referred to this observation in the revised manuscript, stating (line 285) “Widespread super-additivity between developmental enhancers might explain the predominance of developmental genes among genes with the highest transcription rates (Supplementary Fig. 3g) and enable rapid gene induction after signaling and during development”. Regarding housekeeping genes, we now write (line 339) “additive interactions seem sufficient to foster steady transcription of housekeeping genes. However, such interactions still imply that housekeeping enhancers might boost each other, which could explain why housekeeping genes and enhancers tend to form clusters along the Drosophila genome, an arrangement that has previously been shown to be important for their proper transcription [15].”

8. The total number of wildtype enhancers (600 developmental enhancers and 100 housekeeping enhancers) is relatively small compared to over 50,000 developmental enhancers genome-wide (Kvon et al. Nature 2014). There are 2 main questions need to be addressed. 1). If these enhancers activate their gene targets synergistically or additively in their native genomic loci? 2). If the synergistic and additive modes of activation hold true at genome-wide scale? The authors should at least investigate the correlation of the 700 cloned enhancers and their possible pairs in the genomic loci to the endogenous expression level of their gene targets in S2 cells, to see if they fit into the same modes of cooperation achieved from STARR-seq data. In principle, this could be also investigated at genome-wide scale, using mathematic modeling or machine learning approach.

We agree with the reviewer that understanding enhancer cooperativity and target gene regulation *in situ* to predict gene expression at the genome-wide scale would be the ultimate goal of this research field. Our work contributes to this overarching goal by establishing the rules of enhancer-enhancer

cooperativity under standardized conditions, as such rules would otherwise not be experimentally tractable. As the reviewer surely appreciates, the application of these rules to mathematical models of genome-wide gene expression prediction is currently not possible in a meaningful and credible fashion because there are too many unknown parameters (e.g. different promoter sequences with various saturation levels, highly variable distances, influence of chromatin states, regulatory redundancies, etc.).

We now add the overarching goal to predict gene expression and the specific contribution of our effort to the discussion and thank the reviewer for the opportunity to provide additional context to this work. Specifically, we write (line 343): “Future studies should aim at integrating the basic modes of cooperativity between active enhancers that we uncovered here with further regulatory information (e.g. enhancer-promoter distance, CP selectivity, CP saturation) and chromatin states towards the overarching goal of achieving genome-wide predictions of gene activity”.

9. The authors design and cloned several Inducible enhancers and OSC-specific enhancers; however, no relevant results, findings, and discussions are presented. The same experiments should be done at least on subsets of relevant enhancers under inducible conditions (Ecdysone and Heatshock) on S2 cells, and on OSCs, to see if they fit the same synergistic/multiplicative or additive mode.

We had originally included inducible and OSC-specific enhancers as additional negative controls (they are not active in unperturbed S2 cells), to diversify the types of negative control sequences. However, we agree with the reviewer that it would be interesting to extend the scope of this study and examine the modes of cooperativity of inducible enhancers. We thus now conducted STARR-seq in S2 cells induced with ecdysone and in OSC cells and found that both hormone-inducible and OSC-specific developmental enhancers “also combined super-additively (Fig. 2f-g, Supplementary Fig. 2k-l), indicating that this is a common feature of Drosophila developmental enhancers” (line186):

Supplementary Figure 2j: Activity of ecdysone-inducible (left) and OSC-specific (right) enhancers in untreated S2 cells (grey), ecdysone treated S2 cells (blue) and OSC cells (green). Two-sided Wilcoxon test P-values are shown.

Figure 2 f-g: Predicted additive (x axis) versus observed combined activities for ecdysone-inducible enhancers and OSC-specific enhancers (in red) in ecdysone-treated S2 cells (f) or OSC cells (g). As a reference, a subset of pairs containing enhancers that were also active in S2 cells are shown (see color legend). Dotted lines depict the identity lines (where observed combined activities equal expected additive outcomes) and plain lines represent fitted multiplicative models.

Supplementary Fig. 2 k-l: k- The four categories of enhancer pairs shown in Fig. 2f (S2 cells + ecdysone) are plotted separately (scatterplots on the left), and the residuals of the additive model (dotted lines) are quantified (boxplots on the right). Two-sided Wilcoxon test P-values are shown. l- The four categories of enhancer pairs shown in Fig. 2g (OSC cells) are plotted separately (scatterplots on the left), and the residuals of the additive model (dotted lines) are quantified (boxplots on the right). Two-sided Wilcoxon test P-values are shown).

Minor points:

1) The limitation of method used in this work should be addressed.

In the revised version of the manuscript, we now discuss the inherent limitations of episomal assays such as STARR-seq, that are essential for regulatory element testing at scale. We now state that “further studies would be needed to tackle [...] whether other chromatin-related features might further constrain enhancer cooperativity *in situ* [12], and how motif binding affinity might influence these interactions [8]” (line 334). We also acknowledge that “Future studies should aim at integrating the basic modes of cooperativity between active enhancers that we uncovered here with further regulatory information (e.g. enhancer-promoter distance, CP selectivity, CP saturation) and chromatin states towards the overarching goal of achieving genome-wide predictions of gene activity” (line 343). We also specify that super-additivity is largely promiscuous “in our system” (line 233) and that “the *Drosophila* genomic enhancer sequences we have been using typically already contain homotypic and heterotypic combinations of motifs, and future studies could use synthetic sequences to more specifically assess the impact of each motif” (line 308).

2) Page 5, line 144, multiplicative model was more accurate for “88%”, rather than 90%, if written in a precise way.

Thanks for highlighting this mistake, which we fixed (line 156).

3) Line 501-505, the list of selected motifs is not included in mentioned table/column.

We now provide the full list of selected motifs and corresponding PWMs in Supplementary Table 8.

4) Line 546, starts with a typo “l”.

We fixed this typo.

5) Fig. 3c, 3d, “twist” should be “Twist”. In the legend, “In situ” should be italic and “i-n pink” contains typo.

We fixed these typos.

Reviewer #3 (Remarks to the Author):

Genes are often regulated by more than one enhancer, however how these enhancers interact in different biological contexts to coordinate gene expression is an open area of investigation. In this manuscript, Loubiere et al. screen a STARR-seq library of individual and paired enhancers for activity in *Drosophila* S2 cells to examine how developmental enhancers and housekeeping gene enhancers interact to regulate gene expression. The authors' expertise in creating MPRA libraries is highlighted by a clever fusion PCR-based approach to create their paired enhancer STARR-Seq library. This library used a developmental core promoter and was composed of enhancers paired with inert controls (individual enhancers) and enhancer-enhancer pairs, which allowed the authors to compare reporter gene expression in both contexts. Through modeling of their STARR-Seq activity data, the authors find that most developmental enhancers interact in a synergistic manner, while most housekeeping enhancers interact additively in S2 cells. The authors conclude that the synergistic interaction between developmental enhancers is promiscuous in S2 cells, meaning that it is not dependent on specific transcription factor motifs, genomic location, or even promoter used. However, the authors do note that developmental enhancers are more enriched in binding sites for transcription factors with intrinsically disordered regions. This manuscript is generally scientifically sound, but I believe the following major and minor comments should be addressed to provide evidence for the authors' claims and to provide clarity for readers.

We thank the reviewer for their valuable feedback and overall appreciation of our work. We hope that the following point-by-point response and the corresponding revision of the manuscript will clarify all pending concerns.

Major Comments/Concerns:

1. While S2 cells are a great starting point for examining this biological question, it is difficult to believe the authors generalized claims about interactions of developmental enhancers outside of a developmental context, where exposure to different activating and repressive TFs is varied across time and space. For example, in the authors cited reference 10, Bothma et al. state that the hunchback developmental enhancer behaves differently (additively vs sub-additively) depending on the cellular context (levels of Bicoid); this is somewhat captured in the Loubiere et al. STARR-Seq data showing CP saturation when a strong enhancer is used. Additionally, in citation 14, Lam et al. describe how two enhancers for the proopiomelanocortin gene in mice initially act synergistically in embryos and as development progresses to adulthood, these enhancers behave additively. Without some type of developmental biological variable used (different cell lines, synthetic signaling gradients, or a handful of examples using BAC transgene reporters in flies or other system), it is difficult to accept a generalized claim about developmental enhancers with data gathered only from a single cell line.

We thank the reviewer for pointing out that we had not sufficiently clearly explained the use of the term "developmental enhancer". We use the term as in previous work (e.g. Arnold et al., Science 2013; Zabidi et al., Nature 2015) to describe enhancers that regulate developmental and/or cell-type-specific genes as opposed to ubiquitously expressed "housekeeping" genes. We now clarify the use of the term and how rules regarding these enhancers' cooperativity can be derived from single cell lines.

However, we also agree with the reviewer that testing our results with a developmental biological variable would strengthen their generalizability and thank the reviewer for suggesting how to substantially broaden the scope of our work and generalize to a developmental context and a second cell type. We now conducted STARR-seq in S2 cells induced with ecdysone – the main steroid hormone in insects – and in OSC cells, and found that both hormone-inducible and OSC-specific developmental enhancers "also combined super-additively (Fig. 2f-g, Supplementary Fig. 2k-l), indicating that this is a common feature of *Drosophila* developmental enhancers" (see line 186 and corresponding figures below). Many thanks.

Supplementary Figure 2j: Activity of ecdysone-inducible (left) and OSC-specific (right) enhancers in untreated S2 cells (grey), ecdysone treated S2 cells (blue) and OSC cells (green). Two-sided Wilcoxon test P-values are shown.

Figure 2 f-g: Predicted additive (x axis) versus observed combined activities for ecdysone-inducible enhancers and OSC-specific enhancers (in red) in ecdysone-treated S2 cells (f) or OSC cells (g). As a reference, a subset of pairs containing enhancers that were also active in S2 cells are shown (see color legend). Dotted lines depict the identity lines (where observed combined activities equal expected additive outcomes) and plain lines represent fitted multiplicative models.

Supplementary Fig. 2 k-l: k- The four categories of enhancer pairs shown in Fig. 2f (S2 cells + ecdysone) are plotted separately (scatterplots on the left), and the residuals of the additive model (dotted lines) are quantified (boxplots on the right). Two-sided Wilcoxon test P-values are shown. l- The four categories of enhancer pairs shown in Fig. 2g (OSC cells) are plotted separately (scatterplots on the left), and the residuals of the additive model (dotted lines) are quantified (boxplots on the right). Two-sided Wilcoxon test P-values are shown.

a. It would be nice to see, if the sequences are in the dataset, if enhancers that were previously examined by others behave as previously reported. This might add some weight to the authors claims about developmental enhancers.

We ensured that all enhancers behaved similarly to our previous work in S2 cells and OSCs, with and without ecdysone treatment (see Supplementary Fig. 2j). Regarding enhancers selected by the aforementioned work by Bothma (<https://doi.org/10.7554/eLife.07956.002>) or Perry and colleagues (<https://doi.org/10.1073/pnas.1109873108>) in *Drosophila* embryos, they are associated to developmental TF genes (*snail*, *hunchback* and *knirps*) that are not expressed in S2 cells (with or without ecdysone) or OSC cells (FPKM<0.25 in all cases). Consistently, no STARR-seq peaks were detected near these gene loci in S2 cells, precluding further comparisons:

Figure to reviewer: STARR-Seq tracks screenshot at hunchback (*hb*), knirps (*kni*) and snail (*sna*) loci in S2 cells. As a reference, the transcriptionally active kayak locus (the *Drosophila* homolog of *Fos*) is shown on the right.

2. In your library, how do you account for non-similar genomic spacing? The authors saw no difference between genomically close and distant linked enhancers using a 300bp spacer, but does spacing between the enhancers change the way they interact? The synthetically close spacing may perhaps be more similar for housekeeping enhancers (as noted in the text), but what about developmental enhancers? A test of a handful of enhancers that interact endogenously to examine the 300bp spacing used by the authors and comparing this to endogenous spacing would be a useful control.

We agree with the reviewer that assessing enhancer-enhancer cooperativity with a longer spacer would be an interesting extension of our work. We therefore measured enhancer-enhancer cooperativity using a 2kb spacer, a distance that is similar – and slightly longer – than the median enhancer-enhancer distance within the *Drosophila* genome (Supplementary Fig. 2f; longer spacers of tenth of kbs are unfortunately not possible for technical reasons). This confirmed that developmental enhancers are super-additive within this context, and surpassed their predicted additive outcome (see line 177 and new Supplementary Fig. 2h):

Supplementary Fig. 2h: Scatterplot showing the predicted additive (*x* axis) versus observed (*y* axis) combined activities of enhancer pairs in which both candidate sequences are active. The dotted line depicts the identity line (where observed combined activities equal expected additive outcomes) and the plain line represents the fitted multiplicative model. Residuals (observed-predicted additive) are quantified on the right. *i*- Predicted additive (*x* axis) versus observed (*y* axis) combined activities inferred using luciferase assays for 10 different homotypic enhancer pairs, in which 5' and 3' individual enhancers are the same. The identity line ($y=x$) is shown, corresponding to the additive model.

3. The authors claim that TF motifs likely only play a very minor role, if any, in the synergistic behavior of developmental enhancers. They then go on to test this claim by examining TF motifs present in enhancer pairs with activity beyond what their model predicts (residuals). They mutate Trl and Twist sites in active enhancers that contain them or by adding them into their library of enhancers (inert or active sequences). Lines 379 to 386 in the methods section raise some concerns about the experimental approach, but these could just be points of clarification.

a. It is stated that, “we mutated Twist, Trl and Dref motifs by replacing them with random stretches of nucleotides within a set of active enhancers that contained them.”

i. Were the random sequences consistent for each TF site replaced? It seems like multiple were initially used, but how many were actually selected in the end?

We thank the reviewer for bringing up this point. Indeed, to avoid biasing the analysis towards a single random sequence per motif, we used randomly sampled nucleotide stretches that did not match known TF motifs, and used them to replace the different instances of TF motifs. We now report the number of variants in Supplementary Table 1 and rephrase the methods section to improve clarity (line 494), stating that: “we mutated Twist/Trl/Dref motifs in a subset of active enhancers that contained them, by

replacing each motif instance with randomly sampled stretches of nucleotides (containing no known motifs). [...] for each tested condition, the final number of variants can be found in Supplementary Table 1”.

ii. Were the random sequences that were used as “mutated sites” tested for activity or checked to ensure that a new binding site was not created?

Thanks for this question. Indeed, we tested all sequences using the DeepSTARR deep-learning model, which detects known motifs and interactions between them and allows the prediction of enhancer activity from the DNA sequence (see ref. 17., de Almeida et al., Nature genetics, 2018; <https://doi.org/10.1038/s41588-022-01048-5>). This enabled us to select motif/enhancer variants that had a minimal impact on individual enhancers’ activities and thus the specific assessment of enhancer-enhancer cooperativity. We refined our methods section to better explain this point (line 496): “Because we are specifically interested in cooperativity between active enhancer pairs, we needed to avoid mutations that would alter the activity of the individual enhancers, by e.g. creating new motifs and/or deleting essential ones, as this would confound the analysis. In other words, our goal was to preserve the activity of the individual enhancers, while changing the TF motifs we suspected to influence cooperativity. To do so, we started from a large pool of WT sequences and generated 1,000 possible enhancer variants for each condition, changing the position of pasted motifs and/or the stretches of nucleotides being used to replace each motif instance. Then, we predicted the activity of all enhancer variants using DeepSTARR17 and retained the ones with the smallest impact on predicted individual activities”.

b. Lines 381 through 386 detail that many sequence iterations were used per mutation or site addition and those predicted to have the least impact on individual enhancer activity were selected for analysis.

i. This is very confusing for the reader as it seems the data that was chosen to be analyzed was biased towards the null hypothesis for these TFs having an impact on enhancer interaction. Please clarify or explain why this approach was done over using the median or some type of averaging.

Following up on point 3.a.ii., our goal was actually to identify differences in combined activities that cannot be explained/predicted from the individual activities of the two enhancers in the pair. Thus, selecting enhancer variants with similar individual activities is intended and does not bias the analysis towards our null hypothesis (which might for example be the case if we chose candidates based on predicted combined activities). We hope that the revised version of the Methods is now clearer (see point 3.a.ii and line 496).

4. The authors show a small glimpse that binding sites for IDR TFs could be driving developmental enhancers to behave synergistically compared to housekeeping enhancers. A simple experiment here is to add these sites into housekeeping enhancers and use qPCR or a Luciferase assay to see if it causes shifts to a more synergistic interaction. Wrapping the story up with a mechanism would bolster the impact of this manuscript.

To tackle this interesting point, we analyzed enhancers that can activate the housekeeping and the developmental CP and contain some developmental TF motifs with long IDRs, such as Trl and Twist motifs (“dual enhancers”; see ref. 4, Zabidi et al.) and new Supplementary Fig. 3m:

Supplementary Fig. 3m-n: m- Distribution of Trl (top) Twist (middle) or summed (Trl+Twist) motif counts (x axis) within dual enhancers (in yellow) and an activity-matched set of canonical housekeeping enhancers (orange). n- Expected additive

and observed activities of housekeeping (Hk./Hk.) and dual (Dual/Dual) enhancer pairs (x axis) using the RpS12 housekeeping CP. Two-sided Wilcoxon test P-values are shown.

Consistent with our hypothesis and the reviewers' suggestion, we found that dual enhancers with Trl and/or Twist motifs “combine super-additively towards a housekeeping CP, contrasting with the additive behavior of canonical housekeeping enhancers” (see line 333).

Minor Comments/Clarifications:

1. While the authors claim that specific TF binding sites largely don't impact the behavior of enhancer interactions as a whole, could something like binding site affinity (similar to levels of Bicoid in Bothma et al., 2015) impact behavior?

We agree that other variables such as binding site affinity, chromatin, etc. might further influence additive/super-additive behaviors *in situ*, which we now acknowledge line 334: “further studies would be needed to tackle how additive versus super-additive behaviors are encoded at the protein level, and whether other chromatin-related features might further constrain enhancer cooperativity *in situ*[12], and how motif binding affinity might influence these interactions[8]”.

2. Lines 200 through 203:

“Consistently, a LASSO regression using motif counts as input was able to predict the overall activity of enhancer pairs ($R^2=0.37$) but not the residuals ($R^2=0.08$, Supplementary Fig. 3a-b), confirming the association between TF motifs and enhancer activities and suggesting that residuals or synergies do not rely on specific DNA binding motifs.”

i. With the low R^2 value, I don't agree with the statement the LASSO was able to predict enhancer pair activity, thus I don't believe the authors should use this to suggest anything about TF motifs and enhancer activities or their impact on residuals.

We agree that the R^2 values don't correspond to accurate predictions and thank the reviewer for pointing out that our intentions had not been sufficiently clear: we wanted to highlight the difference between the R^2 for predicting the activity (0.37) versus the residuals of the fitted multiplicative model (0.08), i.e. the fact that LASSO achieves 4.6-times higher R^2 s for activity than for residuals.

After re-training these LASSO models using a refined selection of motifs (to include ecdysone-related and OSC-specific motifs required for the new datasets), we now used a fairer way to calculate R^2 values, by only using enhancer pairs that were not used for training across 9 fold cross-validations (0.35 ± 0.04 for activity and 0.06 ± 0.01) The updated manuscript now reads (line 222): “Consistent with the absence of a clear association between specific motifs and strongly enhanced/decreased super-additivity, a LASSO regression using motif counts as input performed poorly at predicting the residuals of the multiplicative model in untreated S2 cells, ecdysone treated S2 cells and OSC cells ($R^2 \approx 0.06\pm0.01$ on held out test sets, see Supplementary Fig. 3b). In contrast, such model performed substantially better at predicting the activity of enhancer pairs in S2 cells ($R^2=0.35\pm0.04$, Supplementary Fig. 3c), and unambiguously identified the motifs that are known to support the activity of S2 enhancers [17], ecdysone-inducible [19] and OSC-specific enhancers [4] (Supplementary Fig. 3d)”.

Supplementary Fig. 3c-d: b- Residuals of fitted multiplicative models with interaction term (y axis) as a function of LASSO predictions (x axis), using a representative set of Drosophila DNA binding motifs (n= 120) in untreated S2 cells (left), ecdysone-treated S2 cells (center) and OSC cells (right). Mean adjusted R-squared (R^2) \pm standard deviation (sd) across 9 different cross-validation folds are shown. c- Combined activities (y axis) as a function of LASSO activity predictions (x axis),

using a representative set of Drosophila DNA binding motifs (n= 120). Mean adjusted R-squared (R²) ± standard deviation (sd) across 9 different cross-validation folds is shown on the top left. d- Lasso coefficients of the top DNA binding motifs identified by LASSO regression in untreated S2 cells (S2 – ecd.), ecdysone-treated S2 cells (S2 + ecd.) and OSC cells (x axis). The sign of the coefficients indicates positive or negative associations.

3. Line 352 “flanked by PCR primers” should be “flanked by PCR primer binding sites”.

We fixed this issue.

4. Line 438 “indexe” should be “index”.

We fixed this typo.

5. Line 545 “tree” should be “three”.

We fixed this typo, thanks.

6. Figure 2D legend: “Boxplot showing, for all enhancer pairs, expected (right) and observed activity values (left) using the three different models.”

a. I believe left and right are meant to be swapped.

Fixed (now in panel 2e), thank you for pointing it to us.

7. Supplementary Figure 3D is mislabeled as C in the legend and is missing statistical test information.

We now added the statistical tests and fixed the labelling.

Reviewers' Comments:

Reviewer #1:

Remarks to the Author:

As stated in my previous review, I think that the study is exciting, elegant, and technically robust. The topic of whether enhancers act additively, super-additively, or sub-additively is one that is of great interest, and the finding that different types of enhancers combine differently is very interesting.

The revised manuscript addresses all of my previous questions. The new Fig 4c is great.

Reviewer #2:

Remarks to the Author:

The revised manuscript by Loubiere et al. is greatly improved and has rigorously addressed the major points raised.

Thorough QC and comparative new experiments and analyses were done to measure/model the activities of single-/combined-, N-terminal/C-terminal enhancers with/without certain TF motifs. However, given by the limited number of CP, TF motifs used, cautions must be born in mind to draw relevant conclusions. Authors should: 1) dedicate a section to discuss the limitations (and further advantages if there are) of the STARR-seq reporter assay on S2 cells in assessing the enhancer pairs coactivity; 2) refrain from generalizing the conclusion drawn from in vitro S2 cells as a in situ features of drosophila development process, particularly in the discussion part.

Other minor points:

1. double check Fig. 3c. Trl motif, the left scatterplot seems inconsistent to the right statistic bar plot, in terms of “Although mutant enhancer pairs remained super-additive, they showed slightly but significantly decreased super-additivity compared to their wt counterparts”. (Line 217)
2. Fig.3e, Fig. legend, the distance of <20kb (or ≤ 20 kb), and >20kb should be stated same as those in Fig. labels and main text.
3. Some supplementary figures should be considered to present as main Figs, e.g. Suppl.Fig. 1c, several plots in Suppl. Fig.3 regarding the analyses of Motif.

Reviewer #3:

Remarks to the Author:

I thank the authors for comprehensively addressing my comments. I am satisfied with the revised manuscript.

POINT-BY-POINT RESPONSE TO REVIEWERS' COMMENTS

Reviewer #1 (Remarks to the Author):

As stated in my previous review, I think that the study is exciting, elegant, and technically robust. The topic of whether enhancers act additively, super-additively, or sub-additively is one that is of great interest, and the finding that different types of enhancers combine differently is very interesting.

The revised manuscript addresses all of my previous questions. The new Fig 4c is great.

We thank the reviewer very much for their enthusiastic feedback and for highlighting the importance of the topic of our work. Many thanks also for the constructive remarks throughout the review process.

Reviewer #1 (Remarks on code availability):

There is a README. I did not attempt to install or run the code.

Reviewer #2 (Remarks to the Author):

The revised manuscript by Loubiere et al. is greatly improved and has rigorously addressed the major points raised.

Thorough QC and comparative new experiments and analyses were done to measure/model the activities of single-/combined-, N-terminal/C-terminal enhancers with/without certain TF motifs. However, given by the limited number of CP, TF motifs used, cautions must be born in mind to draw relevant conclusions.

We thank the reviewer for acknowledging our efforts in revising the manuscript.

Authors should:

1) dedicate a section to discuss the limitations (and further advantages if there are) of the STARR-seq reporter assay on S2 cells in assessing the enhancer pairs coactivity; 2) refrain from generalizing the conclusion drawn from in vitro S2 cells as a in situ features of drosophila development process, particularly in the discussion part.

Thanks for this comment. We now clarified that “While developmental enhancers activating tissue-specific genes are super-additive in our minimal reporter assay, housekeeping enhancers behave additively” (line 283).

Before switching from our MPRA-based results to in-vivo observations, we now stated that “Although STARR-seq in cultured cell lines does not capture all aspects of enhancer cooperativity during development, super-additivity between developmental enhancers might...” (line 289). This initial statement is later developed by highlighting the potential impact of chromatin or binding motifs affinity (line 342), in situ enhancer-enhancer distance and CP saturation (line 351).

When starting to discuss the Trl and Twist motifs, we now emphasized that “while Trl and Twist motifs seem to positively influence super-additivity in our setup” (line 304). This comment is further complemented by our assertion that “future studies could use synthetic sequences to more specifically assess the impact of each motif” (line 318).

Other minor points:

1. double check Fig. 3c. Trl motif, the left scatterplot seems inconsistent to the right statistic bar plot, in terms of “Although mutant enhancer pairs remained super-additive, they showed slightly but significantly decreased super-additivity compared to their wt counterparts”. (Line 217)

We made sure that the box plots correspond to the scatterplot shown in Fig. 3c. We agree that the difference between the two groups is small, and likely driven by the subset of pink points populating the bottom part of the cloud. Nevertheless, it remains significant when using a paired Wilcoxon test to compare wild-type versus mutated variants. This is why we consistently reported that Trl (but also Twist) motifs have a rather limited impact and are not strictly required for super-additivity. To make this point more clear, we refined our abstract to say that “Super-additivity between developmental enhancers is promiscuous and neither depends on the enhancers’ endogenous genomic contexts nor on specific transcription factor motif signatures, but it can be **further** boosted by Twist and Trl motifs” (line 50).

2. Fig.3e, Fig. legend, the distance of <20kb (or \leq 20kb), and >20kb should be stated same as those in Fig. labels and main text.

The legend was fixed. Thank you for pointing out this typo.

3. Some supplementary figures should be considered to present as main Figs, e.g. Suppl.Fig. 1c, several plots in Suppl. Fig.3 regarding the analyses of Motif.

Thank you for highlighting the work shown in the Supplementary Figures. We consider Supplementary Fig. 1c to be a validation experiment, which brings no added value compared to our STARR-seq results. On the other hand, the motif analyses shown in Supplementary Fig. 3 focus on the activity of individual enhancers, which is not the main focus of our manuscript and in fact has been published before (see reference 17). In our case, it serves as a supplementary control. Thank you very much for your understanding.

Reviewer #2 (Remarks on code availability):

The codes contain README file.

Reviewer #3 (Remarks to the Author):

I thank the authors for comprehensively addressing my comments. I am satisfied with the revised manuscript.

We thank the reviewer for his constructive comments during the revision process.